# Machine learning approaches identify immunologic signatures of total and intact HIV DNA during long-term antiretroviral therapy

Lesia Semenova[1], Yingfan Wang[2], Shane Falcinelli[3,4], Nancie Archin[3,5], Alicia D Cooper-Volkheimer[6], David M Margolis[3,4,5], Nilu Goonetilleke[3,4], David M Murdoch[6], Cynthia D Rudin[2]*, Edward P Browne[3,4,5]*

[1]Microsoft Research, Duke University, Durham, United States; [2]Department of Computer Science, Duke University, Durham, United States; [3]UNC HIV Cure Center UNC Chapel Hill, Chapel Hill, United States; [4]Department of Microbiology and Immunology, UNC Chapel Hill, Chapel Hill, United States; [5]Department of Medicine, UNC Chapel Hill, Chapel Hill, United States; [6]Department of Medicine, Duke University, Durham, United States

**\*For correspondence:**
cynthia@cs.duke.edu (CDR);
epbrowne@email.unc.edu (EPB)

**Competing interest:** The authors declare that no competing interests exist.

**Abstract** Understanding the interplay between the HIV reservoir and the host immune system may yield insights into HIV persistence during antiretroviral therapy (ART) and inform strategies for a cure. Here, we applied machine learning (ML) approaches to cross-sectional high-parameter HIV reservoir and immunology data in order to characterize host–reservoir associations and generate new hypotheses about HIV reservoir biology. High-dimensional immunophenotyping, quantification of HIV-specific T cell responses, and measurement of genetically intact and total HIV proviral DNA frequencies were performed on peripheral blood samples from 115 people with HIV (PWH) on long-term ART. Analysis demonstrated that both intact and total proviral DNA frequencies were positively correlated with T cell activation and exhaustion. Years of ART and select bifunctional HIV-specific CD4 T cell responses were negatively correlated with the percentage of intact proviruses. A leave-one-covariate-out inference approach identified specific HIV reservoir and clinical–demographic parameters, such as age and biological sex, that were particularly important in predicting immuno-phenotypes. Overall, immune parameters were more strongly associated with total HIV proviral frequencies than intact proviral frequencies. Uniquely, however, expression of the IL-7 receptor alpha chain (CD127) on CD4 T cells was more strongly correlated with the intact reservoir. Unsupervised dimension reduction analysis identified two main clusters of PWH with distinct immune and reservoir characteristics. Using reservoir correlates identified in these initial analyses, decision tree methods were employed to visualize relationships among multiple immune and clinical–demographic parameters and the HIV reservoir. Finally, using random splits of our data as training-test sets, ML algorithms predicted with approximately 70% accuracy whether a given participant had qualitatively high or low levels of total or intact HIV DNA . The techniques described here may be useful for assessing global patterns within the increasingly high-dimensional data used in HIV reservoir and other studies of complex biology.

## eLife assessment

Semenova et al. have studied a large cross-sectional cohort of people living with HIV on suppressive antiretroviral therapy and performed high dimensional flow-cytometry for analysis with data science/

machine learning approaches to investigate associations of immunological and clinical parameters and intact/total HIV DNA levels (and categorizations). The study is **useful** in introducing these new methods and large data set and appears mostly **solid**, though some of the claims were **incompletely** supported by the modeling results. The authors have revised the text to fairly reflect their results, yet open questions remain about utility, particularly as to the value of categorical classification (vs continuous measurement) of reservoir size.

## Introduction

The advent of antiretroviral therapy (ART) has significantly decreased morbidity and mortality associated with HIV infection. However, a long-lived proviral reservoir precludes a cure of infection (*Finzi et al., 1997*; *Chun et al., 1997*; *Chun et al., 1998*; *Wong et al., 1997*). While a number of cell types have been proposed as contributing to the HIV reservoir, the largest and most well-characterized reservoir is within CD4 T cells. Molecular analyses of residual HIV proviruses in CD4 T cells during ART have shown that the reservoir is comprised of a heterogeneous mixture of full-length replication-competent proviruses and a larger set of defective proviruses containing internal deletions or APOBEC-mediated hypermutation(s) (*Ho et al., 2013*; *Imamichi et al., 2016*; *Bruner et al., 2016*). Infected cells can undergo clonal expansion and contraction over time in response to homeostatic cues or antigen-driven activation (*Maldarelli et al., 2014*; *Wang et al., 2018*; *Collora et al., 2022*). In addition to the presence of a latent HIV reservoir, people with HIV (PWH) on therapy exhibit a persistent level of T cell immune activation and other altered phenotypes compared to uninfected individuals, involving cells of both the innate and adaptive immune systems (*Sponaugle et al., 2023*). Specifically, HIV infection is associated with increased levels of inflammatory biomarkers and increased immune cell activation (*Hunt et al., 2003*; *Bastard et al., 2012*; *Kottilil et al., 2006*; *Lichtfuss et al., 2011*). PWH also exhibit a depleted naive T cell compartment (*Prescott, 1995*) and increased expression of activation/exhaustion and senescence markers such as CD38, PD-1, and KLRG1 (*Day et al., 2006*; *Ibegbu et al., 2005*; *Tavenier et al., 2015*; *Wang et al., 2020*; *Hatano et al., 2013*). The molecular mechanisms underlying persistent immune activation and dysfunction for PWH on ART are unclear, but it is possible that ongoing viral gene expression and reverse transcription generates ligands for pattern recognition receptors in the innate immune system (*Alter et al., 2007*). However, another important potential cause of ongoing immune activation during ART is persistent sequelae of the massive disruptions to T cell homeostasis, intestinal permeability, and lymphoid structures that occur during untreated HIV infection (*Lederman et al., 2013*; *Heather et al., 2015*; *Hunt et al., 2014*).

It is possible that the HIV reservoir and the host immune system interact in multiple complex ways during therapy. The dynamics of how the HIV reservoir affects the immune system and vice versa are important to characterize and may lead to novel ideas to promote reservoir depletion or to inhibit immune dysfunction that may be related to ongoing HIV expression in persistently infected cells. A number of previous studies have examined the relationship between reservoir size and the immune system (*Horsburgh et al., 2020*; *Chomont et al., 2009*; *Cockerham et al., 2014*; *Hatano et al., 2013*; *Ruggiero et al., 2015*; *Lee et al., 2019*; *Huang et al., 2023*; *Banga et al., 2016*; *Fromentin et al., 2019*; *Fromentin et al., 2016*; *Pardons et al., 2019*; *Dufour et al., 2023*; *Zhu et al., 2023*; *Wu et al., 2023*; *Gálvez et al., 2021*; *Bernal et al., 2023*; *Astorga-Gamaza et al., 2023*; *Fisher et al., 2023*; *Dubé et al., 2023*; *Takata et al., 2023*). Consistent major findings from these studies include enrichment of the HIV reservoir in CD4 T cells expressing activation and exhaustion markers, positive correlation of the reservoir with central memory T cells (Tcm) and transitional memory (Ttm) T cell subsets, and inverse correlation of the HIV reservoir with the CD4 nadir and the CD4/CD8 T cell ratio. Recent studies indicate enrichment of the intact HIV reservoir in cells expressing VLA-4, but no correlation with classic T cell activation markers (*Dufour et al., 2023*; *Horsburgh et al., 2020*). One important limitation of most of these studies is that they were not able to distinguish intact proviruses from defective proviruses, making the relationship between the immune system and the intact reservoir unclear. Furthermore, most of these studies examined a limited number of surface markers and may have missed associations that would be revealed by a higher-resolution examination of immunophenotypes. Additional assessment of immune correlates of the intact HIV reservoir in additional cohorts is thus needed. Finally, there is a recent, growing appreciation for the profound influence of ART initiation timing and length of treatment on the size and composition of the HIV reservoir that

**Table 1.** Participant demographic and clinical characteristics.

For demographics and clinical information, we report percentage for categorical variables, medians, and [Q1, Q3] for real-value variables. ART is antiretroviral therapy. CD4 counts reported in cells/mm³. Years of HIV has 1 missing value, years of ART has 7, and CD4 nadir has 3; consequently, these missing values are not included in median and quantiles computations. Years before ART means years of HIV infection before ART initiation.

| | Percentage (count) | Median | [Q1, Q3] | [Min, Max] |
|---|---|---|---|---|
| Age | | 45 | [37, 53] | [23, 65] |
| Sex (% male) | 76.52% (88) | | | |
| Race | | | | |
| Black | 60% (69) | | | |
| White | 37.39% (43) | | | |
| Other | 2.61% (3) | | | |
| Years of HIV | | 11 | [7, 19.85] | [1, 33.6] |
| Years before ART<1 | 55.65% (64) | | | |
| Years before ART≥1 | 38.26% (44) | | | |
| Years before ART = NA | 6.09% (7) | | | |
| Years of ART | | 9 | [5.23, 16.63] | [0.9,33.5] |
| Recent CD4 count | | 799 | [624.5, 962] | [319, 1970] |
| CD4 nadir | | 313.5 | [163.25, 463.25] | [2, 1080] |

must be accounted for in assessing persistent immune activation in PWH on therapy (*Siliciano and Siliciano, 2022*; *Sponaugle et al., 2023*).

In this study, we use a mixture of traditional statistical approaches and novel data science methods to assess the relationship between the immune system and the HIV reservoir across a cross-sectional study of 115 PWH. We identify numerous significant correlations between immune cell populations and the size of the total HIV reservoir, as well as smaller number of associations with the intact reservoir, including CD127 expression in CD4 T cells. This study highlights the potential of machine learning (ML) approaches to visualize global patterns in high-parameter studies of the HIV reservoir. Additionally, these approaches corroborate recent findings in the HIV persistence field regarding preferential intact proviral decay and immune dynamics and highlight the complex immunologic signatures of HIV latency.

## Results

### Cohort description, reservoir quantification, and immunophenotyping

This cross-sectional study evaluated 115 participants with HIV infection for at least 1 year and who had been receiving ART for at least 0.9 years (median 9 years, interquartile range [IQR] 5.2–16.6) (*Table 1*). The study cohort was 77% male, 60% black, and median participant age was 45 years.

The intact proviral DNA assay (IPDA) was performed on isolated total CD4 T cells (*Bruner et al., 2019*). The IPDA estimates of the frequency of total proviruses, intact proviruses, and the percent intact of total proviruses for each participant. Immunophenotyping was performed using 25-color spectral flow cytometry on peripheral blood mononuclear cells (PBMCs) from each study participant. A representative flow gating is shown in *Figure 1—figure supplement 1*. Briefly, cells within the live lymphocytes gate were first defined as T cells (CD3+/CD56−), NK cells (CD3−/CD56+), or NKT cells (CD3+/CD56+). T cells were then further subdivided as CD4+ or CD8+ T cells, and then as naive T cells (Tn, CD45RA+/CCR7+), central memory T cells (Tcm, CD45RA−/CCR7+), effector memory cells (Tem, CD45RA−/CCR7−), and terminally differentiated effector cells (Teff, CD45RA+/CCR7−). Expression of various surface markers within each of these subsets was then examined (CD38, HLA-DR, PD-1, KLRG1, CD127, CD27, NKG2A). In addition to measuring the baseline abundance of surface proteins, we also stimulated PBMCs from each participant with or without a pool of HIV-derived

peptides and measured intracellular cytokines Tumor necrosis factor alpha (TNFα), Interleukin-2 (IL-2), and Interferon-gamma (IFNγ) as well as the surface expression of the degranulation marker CD107a. HIV-specific T cell responses were defined as the difference in frequency of cytokine/CD107a+ cells between HIV stimulated and nonstimuated cells. Frequencies of CD4 or CD8 T cells that were single or double positive for TNFα, IL-2, IFNγ, or CD107a were used for analysis.

These analyses resulted in the determination of a total of 144 parameters for the cohort of $n =$ 115 PWH on ART. The 144 parameters include 133 immunophenotypic cell frequencies, 3 HIV reservoir parameters (frequency of intact HIV DNA, frequency of total HIV DNA, and percentage intact of total HIV DNA), and 8 clinical–demographic parameters. Clinical–demographic variables (*Table 1*) included age, biological sex, years of ART, estimated years of HIV infection prior to ART (here, we use categorical variables: years before ART = NA, years before ART < 1, years before ART $\geq$ 1), CD4 nadir, most recent CD4 T cell count, and race (Caucasian, African-American, other). No significant differences in the reservoir based on biological sex or race were observed (*Supplementary file 1a*). The median and IQR for all immune cells population variables are presented in *Supplementary file 1b*.

## Correlation of total, intact, and percent intact HIV DNA with immunophenotypes

For the initial analysis, Spearman correlation of the clinical–demographic and immunophenotypic data with HIV reservoir metrics was assessed. 69 host variables correlated (unadjusted p < 0.05) with one or more characteristics of the HIV reservoir (*Supplementary file 1c*). Following Benjamini–Hochberg p-value adjustment for multiple comparisons using a false discovery rate correction at 5%, 31 variables were found to be correlated with total HIV DNA, 3 with intact HIV DNA, and 4 with the percentage of intact HIV DNA (*Table 2*). Of note, the correlations observed were generally weak to moderate in magnitude (the mean of the absolute values of Spearman correlation coefficients was 0.31 for total HIV DNA, 0.33 for intact HIV DNA, and 0.35 for percent intact HIV DNA).

With regard to clinical–demographic variables, CD4 nadir prior to ART initiation was negatively correlated with total HIV DNA (Spearman $r = -0.32$), whereas age ($r = 0.31$) and years of ART ($r = 0.31$) were positively correlated with total HIV DNA (*Table 2*). Years of ART ($r = -0.45$) was negatively correlated with the percentage of intact proviruses (*Table 2*), consistent with the hypothesis that intact proviruses are progressively eliminated from the reservoir over time. We then examined the association between the reservoir and time on ART further (*Figure 1*). Correlation plots of total reservoir frequency, intact reservoir frequency, and percentage intact HIV DNA versus years of ART were visualized (*Figure 1A–C*). The plots showed no significant correlation for intact HIV DNA frequency versus years of ART (*Figure 1B*), while total HIV DNA reservoir frequency was positively correlated with the length of ART treatment (*Figure 1A*, Spearman $r = 0.31$). When we examined a plot of percent intact proviruses versus time on therapy (*Figure 1C*), we observed a biphasic decay pattern, consistent with previous reports (*Peluso et al., 2020*; *Gandhi et al., 2023*; *McMyn et al., 2023*). Indeed, when we fitted piece-wise linear functions with allowances for up to two breaks, the best-fit model had decay slope changes at 1.6 and 9 years of ART. The $R^2$ score of a piece-wise linear model with two breaks was 44.18%, while the score for a linear model without any breaks was 16.3%. In PWH within approximately 0–6 years of ART initiation, a significant proportion of the participants exhibited high fractions of intact proviruses (50–90%). After approximately 6 years of ART, however, the majority of participants had reservoirs for which intact proviruses represented a minor fraction of the overall reservoir (0–30%). Of note, several immune features also demonstrated statistically significant correlations with years of ART, including CD8 T cell expression of CD38 (negative correlation), CD8 Tcm frequency (positive correlation), and CD107a+IL-2+IFNγ−TNFα− CD8 T cells (positive correlation) (*Figure 1—figure supplement 2*, *Figure 1—figure supplement 3*, *Figure 1—figure supplement 4*, *Supplementary file 1d*).

Examination of immunophenotypic parameter correlations with HIV DNA metrics demonstrated several notable correlations (*Table 2*, *Figure 1—figure supplement 2*, *Figure 1—figure supplement 3*, *Figure 1—figure supplement 4*). As previously reported, CD4 T cell frequencies were negatively correlated with total HIV DNA frequencies ($r = -0.36$), whereas CD8 T cells were positively correlated ($r = 0.41$). This relationship held true for intact HIV DNA frequencies but not for the percentage of intact HIV DNA (*Table 2*). Frequencies of CD4 and CD8 T naive cells, as well as their expression of CD38, were significantly negatively correlated with total HIV DNA frequency, consistent with the

**Table 2.** PWH features correlate with HIV reservoir characteristics.

The abundance of 144 immune cell populations was determined by flow cytometry and the HIV reservoir was quantified by intact proviral DNA assay for a cohort of 115 people with HIV (PWH). Each abundance and clinical and demographic variable was correlated with total HIV reservoir frequency, intact reservoir frequency, and the percentage of intact proviruses. Spearman correlation coefficients (bold) are shown for 36 variables that had significant p-values (<0.05) after Benjamini–Hochberg correction for multiple comparisons. Each feature/subset is ranked by the absolute value of the correlation coefficient for the total reservoir frequency. For years of ART, we compute correlation based on 108 participants, excluding participants with missing years of ART values.

| Variable | Total | Intact | Percent intact |
|---|---|---|---|
| %CD8 T | **0.4052** | **0.3562** | 0.0068 |
| %CD38+/HLA-DR− CD4 T | **−0.3891** | −0.1098 | 0.2664 |
| %KLRG1−/PD-1− CD4 T | **−0.3808** | −0.1289 | 0.2334 |
| %Tn CD4 T | **−0.3802** | −0.2031 | 0.1714 |
| %NKG2A+ CD4 T | **0.3618** | 0.2904 | 0.0179 |
| %PD-1−/CCR7+ CD4 T | **−0.3590** | −0.1082 | 0.2283 |
| %CD4 T | **−0.3564** | **−0.3195** | −0.0079 |
| %Tcm CD8 T | **0.3466** | 0.1752 | −0.1814 |
| %CD38+ CD4 T | **−0.3366** | −0.0611 | 0.2829 |
| %PD-1−/CCR7− CD4 T | **0.3300** | 0.1824 | −0.0938 |
| %CD38+/HLA-DR− CD8 T | **−0.3267** | −0.0837 | 0.2636 |
| %PD-1+ CD4 T | **0.3222** | 0.0664 | −0.2470 |
| Age | **0.3172** | 0.1669 | −0.1471 |
| CD4 nadir | **−0.3164** | −0.1512 | 0.1927 |
| %PD-1+ Tn CD4 T | **0.3119** | 0.1246 | −0.0962 |
| %Tn CD8 T | **−0.3028** | −0.2697 | 0.0154 |
| Years of ART | **0.3062** | −0.0688 | **−0.4523** |
| %PD-1+/CCR7+ CD8 T | **0.2926** | 0.1361 | −0.1254 |
| %CD38−/HLA-DR+ CD4 T | **0.2849** | 0.0920 | −0.1500 |
| %KLRG1−/PD-1− CD8 T | **−0.2757** | −0.2182 | 0.0162 |
| %PD-1+ Tn CD8 T | **0.2738** | 0.1411 | −0.0705 |
| %CD38+/HLA-DR− Tn CD8 T | **−0.2676** | −0.0610 | 0.2119 |
| %PD-1+/CCR7+ CD4 T | **0.2665** | 0.0859 | −0.1673 |
| %KLRG1+/CD27+ CD8 T | **0.2606** | 0.2212 | 0.0319 |
| %CD27+ CD4 T | **−0.2602** | −0.0531 | 0.1767 |
| %KLRG1−/CD27+ CD4 T | **−0.2575** | −0.0464 | 0.2171 |
| %HLA-DR+ CD4 T | **0.2565** | 0.1110 | −0.0970 |
| %PD-1+ CD8 T | **0.2541** | 0.1969 | 0.0035 |
| %CD38+ CD8 T | **−0.2418** | 0.0109 | **0.3114** |
| %CD38−/HLA-DR+ Tn CD8 T | **0.2404** | 0.0738 | −0.1346 |
| %Tem CD4 T | **0.2402** | 0.0888 | −0.1669 |
| %CD127+ CD4 T | −0.1298 | **−0.3160** | −0.2539 |
| %CD107a−IFNγ−IL-2+TNFα+ CD4 T | 0.0568 | −0.1775 | **−0.3223** |
| %CD107a−IFNγ+IL-2+TNFα− CD4 T | 0.0285 | −0.2455 | **−0.3265** |

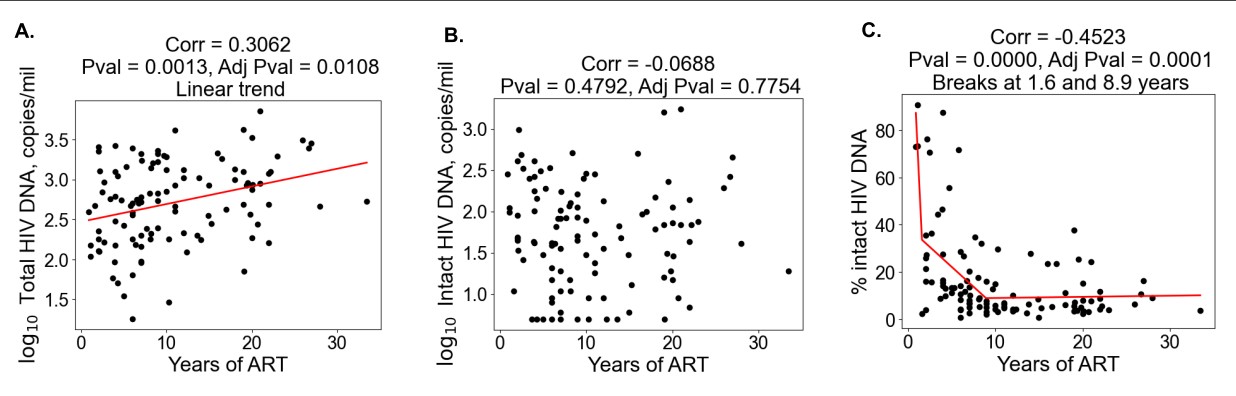

**Figure 1.** Duration of treatment and the HIV reservoir. Scatterplots for years of antiretroviral therapy (ART) versus total HIV reservoir frequency (**A**), intact reservoir frequency (**B**), and percent intact (**C**) are shown. Each dot represents an individual study participant. Correlation coefficients and corresponding p-values are shown for each plot. Participants that have missing values of years of ART were not included in the plot. For percent intact, piece-wise linear function with two breaks is fitted. For total HIV reservoir frequency a linear function is fitted.

The online version of this article includes the following figure supplement(s) for figure 1:

**Figure supplement 1.** Representative flow cytometry gating is shown for one sample from the 115-person cohort.

**Figure supplement 2.** Abundance of immune cell subsets correlates with HIV reservoir (part I).

**Figure supplement 3.** Abundance of immune cell subsets correlates with HIV reservoir (part II).

**Figure supplement 4.** Abundance of immune cell subsets correlates with HIV reservoir (part III).

**Figure supplement 5.** CD4/CD8 and (%CD127+ CD4T)/CD8 ratios correlate with total and intact reservoir frequency.

hypothesis that larger reservoirs are associated with depletion of the naive T cell subsets and increased immune activation. In contrast, CD4 and CD8 Tcm and Tem cells were weakly positively correlated with total HIV DNA, possibly due to increased differentiation of the naive compartment into these memory subsets in people with larger reservoirs. Markers of T cell activation and exhaustion were weakly but significantly positively correlated with total HIV DNA, including NKG2A+ CD4 T, PD-1+ CD4 T, HLA-DR+ CD4 T, PD-1+ CD8 T, and PD-1+/CCR7+ CD8 T frequencies, among others (**Table 2**, **Figure 1—figure supplement 2**, **Figure 1—figure supplement 3**, **Figure 1—figure supplement 4**). For both intact and total HIV DNA, an inverse correlation with the CD4/CD8 T cell ratio was observed (**Figure 1—figure supplement 5**). Interestingly, for the percentage of intact HIV DNA, negative correlations were observed with the frequencies of set subsets of bifunctional HIV peptide-stimulated CD4 T cells (IL-2+TNFα+ and IFN γ+IL-2+), as well as with years on ART. Notably, the magnitude of correlation with immune parameters was typically stronger for total HIV DNA compared to intact HIV DNA (**Table 2**). This pattern may reflect a larger influence of total HIV DNA on immunophenotype of the host, due to the overall higher abundance of total HIV DNA relative to the intact reservoir. A notable exception to this pattern was CD127− CD4 T cell frequencies, which were significantly positively correlated with intact but not total HIV DNA.

Overall these findings support the notion that larger HIV reservoirs are associated with increased levels of immune activation and depletion of the naive T cell compartment. Furthermore, the data indicate that there may be some immune features, such as CD127 expression on CD4 T cells that are uniquely associated with the intact reservoir rather than the total reservoir.

## Leave-one-covariate-out inference identifies specific HIV reservoir and clinical–demographic parameters important for the prediction of immunophenotypes

Given the evidence that reservoir and immune recovery dynamics likely occur in concert, we next used a variable importance approach – LOCO inference analysis, to account for potential confounding variables and more carefully assess the relative importance of a given variable to an immunophenotype. The LOCO inference approach is described in **Figure 2A**. First, a least-squares regression linear model that predicts the dependent variable is fitted on a set of independent variables, and an $R^2$ value is generated (**Figure 2A**, Step 1). Next, one independent variable is excluded from the model,

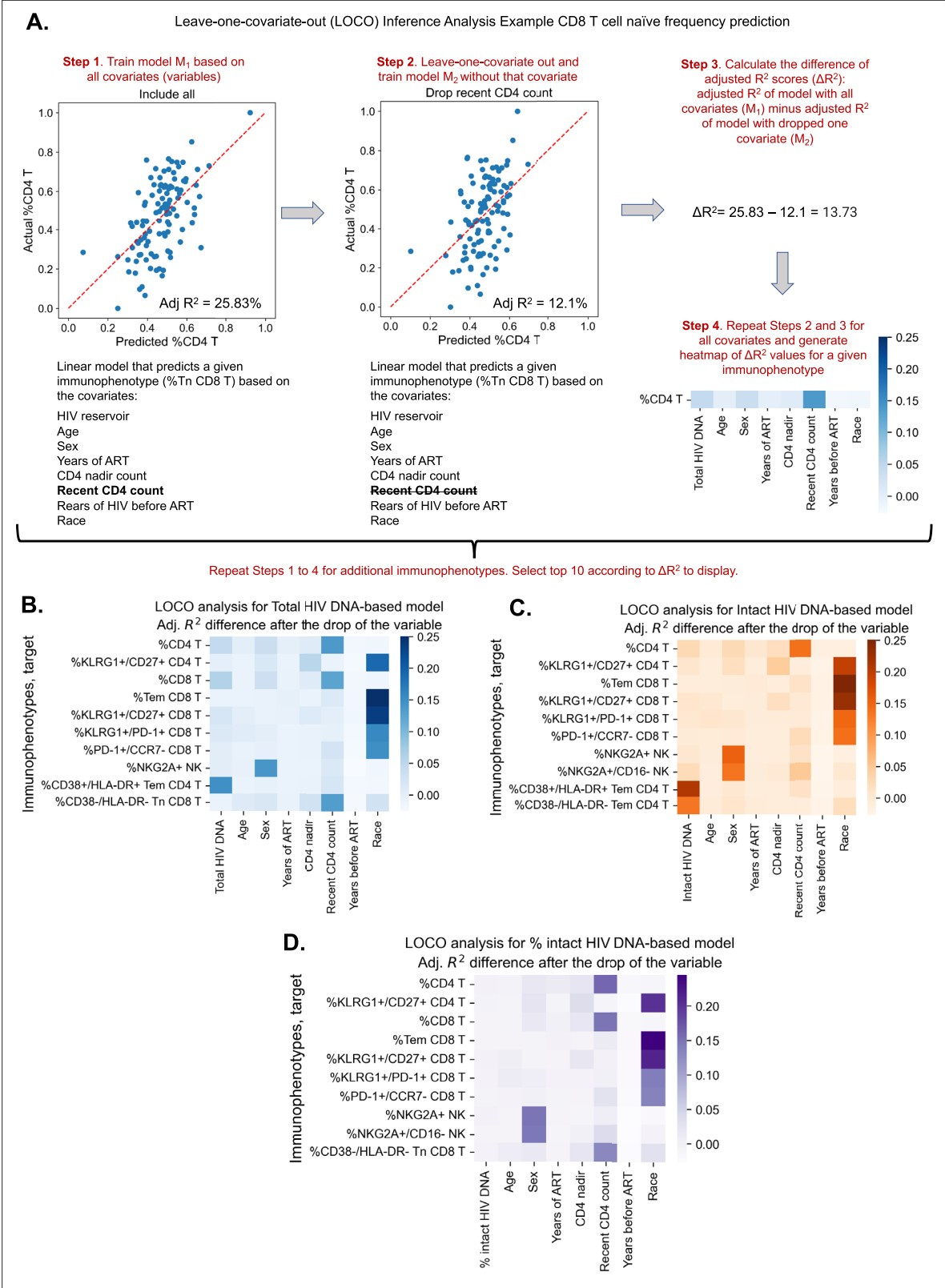

**Figure 2.** Leave-one-covariate-out (LOCO) analysis for clinical–demographic features and reservoir characteristics while predicting immunophenotypes. (**A**) Explanation of LOCO analysis based on example of %CD4 T for clinical–demographic features and reservoir characteristics while predicting immunophenotypes. Analysis was performed for all 133 immunophenotypes considered in the study. The top 10 biggest drops in adjusted $R^2$ scores are reported for models that use total reservoir frequency (**B**), intact reservoir frequency (**C**), or percent intact (**D**) as features in addition to clinical and

*Figure 2 continued on next page*

*Figure 2 continued*

demographic information. Participants with missing years of antiretroviral therapy (ART) values are excluded from this analysis. The missing value of the CD4 nadir for one participant is imputed.

The online version of this article includes the following figure supplement(s) for figure 2:

**Figure supplement 1.** Leave-one-covariate-out (LOCO) analysis visualization for all 133 immunophenotypes.

**Figure supplement 2.** Coefficient visualization for linear regression models that predict immunophenotypes in *Figure 2B–D*.

the model is refitted, and an adjusted $R^2$ value is determined (*Figure 2A*, Step 2). Finally, the change in $R^2$ scores, $\Delta R^2$, is calculated (*Figure 2A*, Step 3). This process is then repeated for all independent variables in the model, and the $\Delta R^2$ values are plotted in a heatmap for the variables of interest (*Figure 2B–D*).

We applied the LOCO inference analysis to all 133 immunophenotype parameters (dependent variables). Three separate analyses were conducted for a set of independent variables consisting of age, sex, years of ART, CD4 nadir, recent CD4 account, years of HIV infection prior to ART (=NA, < 1, ≥ 1), and one of the three HIV reservoir metrics (total HIV DNA, intact HIV DNA, or percentage intact HIV DNA). Multicollinearity analysis confirmed that independent variables from this analysis were not correlated with each other to a degree that would interfere with model generation (*Supplementary file 1c*). Each analysis produced a heatmap of 133 dependent immunophenotype variables, with visualization of the variables for each immunophenotype (*Figure 2—figure supplement 1*, *Supplementary file 1f–h*). For each of these analyses, we displayed the top 10 variables with the largest $\Delta R^2$ value for LOCO inference incorporating either total (*Figure 2B*), intact (*Figure 2C*), or percentage intact (*Figure 2D*) HIV DNA.

The least-squares linear regression models were generally weakly explanatory ($R^2$ values in range [−11.11%, 35.36%]) of the global variability in immunophenotypes (*Supplementary file 1f–h*, *Figure 2B–D*, *Figure 2—figure supplement 2*). Across all three analyses, age and race emerged as an important covariate of several immunophenotypes related to T cell subsets, particularly the naive and central memory compartments, as well as KLRG1, a marker of immune senescence (*Figure 2* and *Figure 2—figure supplement 1*). Interestingly, biological sex was a key variable for model prediction of NK cell NKG2A expression, and race was important for Tem CD8, KLRG1+/CD27+ CD8 frequency for all three models. Recent CD4 count was important for predicting the frequencies of HLA-DR− CD8 Tn cells and CD4, CD8 T cell frequencies, but more so for the models incorporating total HIV DNA and percent intact than the intact HIV DNA frequency (*Figure 2A–C*). Total HIV DNA frequency was an important model input for prediction of PD-1+ CD4 T naive cells, HIV-specific IL-2+TNFα−IFNγ− CD107−CD4 T cells, and CD4/CD8 T cell frequencies, among others (*Figure 2B*, *Figure 2—figure supplement 1*). Total HIV DNA frequency and intact HIV DNA frequency were also important predictors of CD4 T effector memory HLA-DR+CD38+ expression (*Figure 2B, C*).

In general, total and intact HIV DNA were predictive of similar immune features (*Figure 2—figure supplement 1*); however, total HIV DNA generally typically had greater magnitude $\Delta R^2$ than intact HIV DNA. The percentage of intact HIV DNA did not appear to contribute greatly to the predictive power of the model for the variance in the immunophenotypes examined (*Figure 2—figure supplement 1C*), most likely due to its relatively high correlation with years of ART. Overall these findings further demonstrate the existence of a number of clinical parameters that are associated with immune signatures in PWH, including age, gender, and years of ART. Additionally, these results highlight key potential confounding variables that should be considered when interpreting the association of virological parameters with the immune system of PWH.

## Receiver operator characteristic curve analysis demonstrates distinct immune parameters associated with intact and total reservoir frequencies

The complex relationship between the HIV reservoir and the immune system led us to evaluate if ML approaches that combine several parameters into models of the HIV reservoir could be a useful approach to understand this interaction. Before this could be attempted, we first identified a defined set of the most valuable variables from which ML models could be built. To achieve this, we first binarized the three HIV reservoir metrics (total HIV DNA frequency, intact HIV DNA frequency, %intact HIV

DNA) into high (above median) or low (below median) reservoir groups and generated receiver operator (ROC) curves for the 144 clinical–demographic and immunophenotype parameters (*Figure 3*). The area under the curve (AUC) of the ROC curve indicates the ability of the feature (immune parameter or clinical/demographic information) to correctly identifying a participant as having qualitatively low or high total HIV DNA frequency (*Figure 3A*), intact HIV DNA frequency (*Figure 3B*), or percentage intact HIV DNA (*Figure 3C*). A model that randomly guesses high or low reservoir has an AUC of 0.5. In this analysis, 57 host variables had an AUC higher than 0.6 for one or more of total HIV DNA, intact HIV DNA, or percent intact DNA (*Supplementary file 1i*). The most effective individual immune markers for classification of high versus low total HIV DNA frequency included %NKG2A+CD4 T (AUC = 0.70), %PD-1+Tn CD4 T (AUC = 0.68), and %CD38+/HLA-DR− CD8 T (AUC = 0.68). In contrast, the most effective markers for classifying based on the frequency of intact HIV DNA frequency included %CD127+ CD4 T (AUC = 0.71), %CD8 T (AUC = 0.66), and %Tn CD8 T (AUC = 0.65). Finally, for percentage intact HIV DNA, years of ART (AUC = 0.72), %CD107−IFNγ−IL-2+TNFα+ CD4 T (AUC = 0.65) and %KLRG1+/PD1+ CD4 T (AUC = 0.65) were the most effective (*Figure 3*). Similar to the findings in the Spearman correlations (*Table 2*) and LOCO inference analysis (*Figure 2*), there were more variables that had AUC values >0.6 for total reservoir frequency (*Supplementary file 1i*): 44 variables above this threshold for the total reservoir, but only 19 variables for the intact reservoir and 23 variables for percent intact. Overall, this approach allowed us to derive a ranked list of the most predictive immune parameters for each aspect of the HIV reservoir, and these highly ranked features were thus used for subsequent dimension reduction (DR) and ML modeling.

## DR reveals two clusters of PWH with distinct HIV reservoirs

To examine the overall structure of dataset we employed an unsupervised DR machine learning approach (PaCMAP; *Wang et al., 2021*) using the 10 immune cell features with the highest AUC values for classifying participants based on total HIV DNA frequency, intact HIV DNA frequency, or percentage intact HIV DNA (*Figure 3*, *Supplementary file 1i*). Interestingly, while no clear clustering was observed when we use intact reservoir frequency or percentage intact HIV DNA-associated features, we observed two distinct clusters of PWH (clusters 1 and 2, *Figure 4A*) when using total HIV DNA-associated features. These clusters were of roughly equal size with 50 participants in cluster 1 and 65 in cluster 2. Projecting total reservoir frequency (above or below the median) onto the clustering plot, we observe a strong distinction between the two clusters in terms of total HIV DNA reservoir characteristics (*Figure 4B, C*). Analysis of quantitative reservoir frequencies across clusters (*Figure 4C–E*) demonstrated that cluster 1 is characterized by a smaller total HIV reservoir frequency, but greater percentage of intact proviruses (*Figure 4C–E*). In contrast, cluster 2 is defined by a larger total reservoir frequency, but lower percentage of intact proviruses (*Figure 4C–E*). The frequency of intact proviruses also tended to be lower for cluster 1 but this difference was not significant. We also visualized the dataset using principal component analysis (*Figure 4—figure supplement 2*), and observed that participants from each of the clusters identified by PaCMAP occupied different areas of the plot, although the separation was not as clear as with PaCMAP.

To gain insight into the immune cell features that distinguish these two clusters, we generated ROC curves for the 133 immune parameters based on their ability to identify membership in cluster 1 versus cluster 2 for each participant (*Figure 4F*). Many features that distinguished the clusters overlapped with the features selected for the clustering analysis, including expression patterns of CD38, HLA-DR, KLRG1, and PD-1 on CD4 and CD8 T cells and CD4 T naive cell frequencies. However, some novel features emerged, including CD4 KLRG1−CD27+ frequency, CD8 KLRG1−PD1− frequency, and CD8 Tn frequency (*Figure 4F* and *Figure 4—figure supplement 1*). Cluster 2, which had a higher total and intact absolute reservoir frequency but lower percentage of intact proviruses, was associated with a lower fraction of naive CD4 and CD8 T cells. Additionally, cluster 2 had generally higher expression of immune exhaustion markers, including KLRG1 and PD-1 (*Figure 4—figure supplement 1*). These data indicate that cluster membership is highly associated with the overall level of immune activation and exhaustion for the individual. Notably, with regard to other clinical variables, cluster 2 had an older median age, longer time on ART, lower CD4 nadir, and lower current CD4 T cell count (*Figure 4—figure supplement 1*). We have previous shown that cannabis (CB) use is associated with lower levels of immune activation markers in PWH, despite having minimal impact on the size of the HIV reservoir (*Falcinelli et al., 2023*). Interestingly, a subset of participants in this study had clinical data regarding

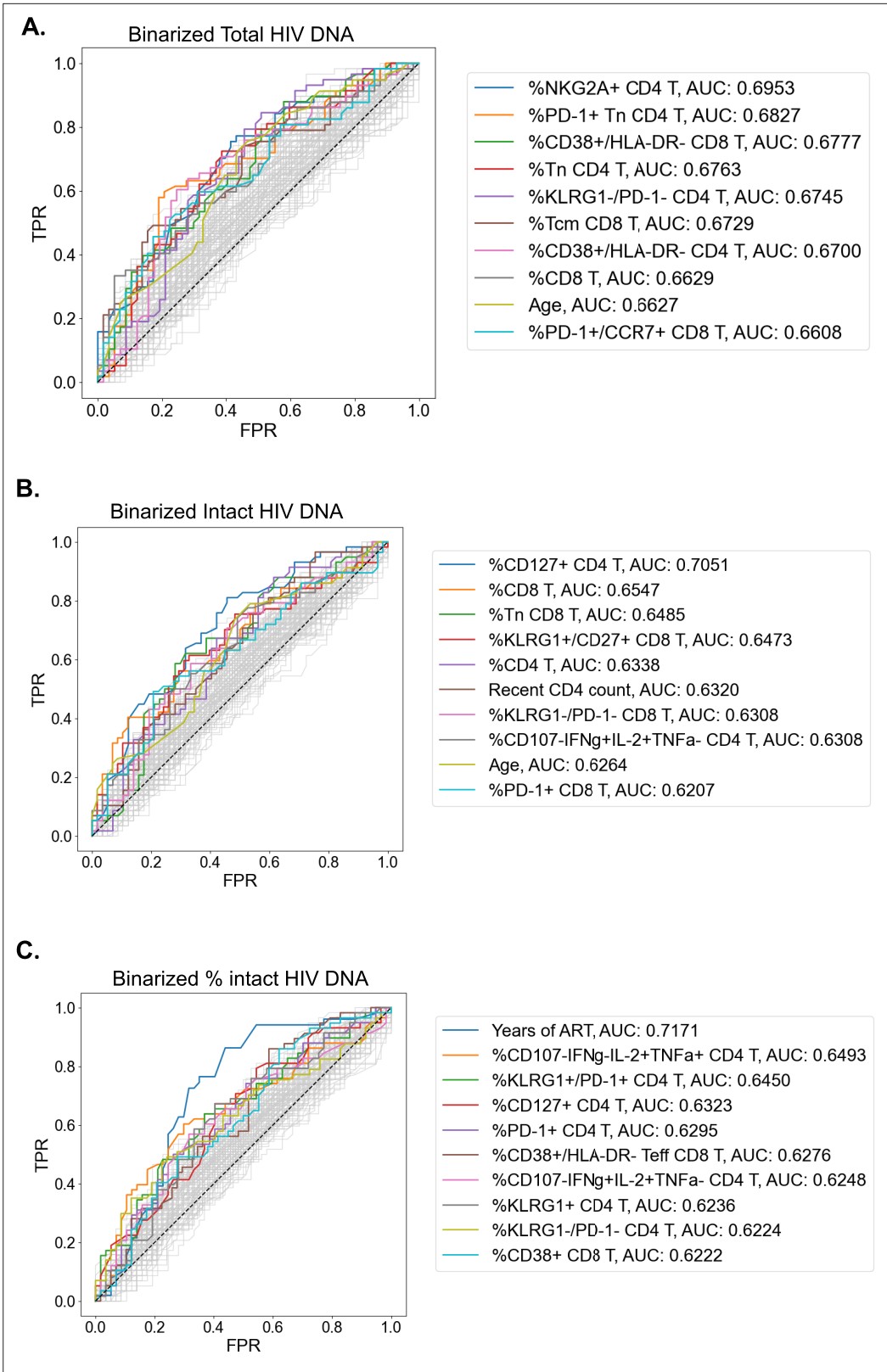

**Figure 3.** Receiver operating characteristic (ROC) curves identify people with HIV (PWH) parameters that can classify reservoir characteristics. For total reservoir frequency (**A**), intact reservoir frequency (**B**), and percent intact (**C**), ROC curves are plotted for all 144 immune markers, demographics, and clinical variables (shown in gray). Axes represent the true positive rate (TPR) and the false positive rate (FPR) for each variable for classifying study

*Figure 3 continued*

participants into low (below median) versus high (above median) reservoir frequency. ROC curves for 10 variables with the highest area under the curve (AUC) values are shown in color for each HIV reservoir characteristic. Striped black lines represent the ROC curves of a random model. For years of antiretroviral therapy (ART) ROC curves, we exclude participants with missing years of ART values.

CB use. CB users were enriched in cluster 1, which had lower levels of immune exhaustion markers (*Figure 4—figure supplement 1*). Overall, these data demonstrate that, within the cohort of PWH on long-term ART, there were two major clusters of participants with distinct reservoir characteristics and immunophenotypes.

## Decision tree visualization of PWH with respect to reservoir characteristics

Since the interaction of the immune system and the HIV reservoir is multifactorial, we hypothesized that models that consider multiple parameters simultaneously could more accurately describe the overall dataset, and provide insights regarding the biology of the HIV reservoir and the host immune system. To accomplish this, we employed a decision tree approach to visualize combinations of variables that classify participants as having high (above median) or low (below median) reservoir frequency. Compared to the DR technique, decision tree visualization is an interpretable supervised approach and does not require post hoc analysis.

We first selected 35 variables with the highest ROC AUC values for either total reservoir frequency, intact reservoir frequency, or percentage intact HIV DNA to be considered for model generation (*Figure 4*). Using these parameters, we fitted Generalized and Scalable Optimal Sparse Decision Trees (GOSDT) (*Lin et al., 2020*) to the data in order to classify high versus low total HIV DNA, intact HIV DNA, or percentage intact HIV DNA. We required the trees to achieve at least 80% accuracy for classifying PWH in the cohort, as well as have at least five PWH in each leaf. Since these trees are based on the entire dataset, these models are thus descriptive rather than predictive.

For the total HIV DNA frequency decision tree, only four immune variables were required to accurately describe high versus low total HIV DNA status (*Figure 5A*, *Figure 5—figure supplement 1*): CD8 T cell frequency, CD4 nadir, %CD38+HLA-DR− CD8 T cells, and %NKG2A+ CD4 T cells. The tree divided the cohort into five subgroups (leaves), among which three have high total reservoir frequency and two have low total reservoir frequency. Comparing the labels provided by GOSDT model with the actual data, the tree achieved 83.5% accuracy (i.e. misclassifying 19 PWH among the overall cohort of 115 PWH). Notably, when we combined all samples from 'high total reservoir' leaves and all samples from 'low total reservoir' leaves, we observed a significant difference in the actual median total reservoir frequencies for these two groups (266/M for low total and 1288.5/M for high total, Mann–Whitney $U$ test p-value is 3.56e−13, *Figure 5B*).

We then repeated this approach for classifying the cohort participants with respect to intact reservoir frequency (*Figure 5C*, *Figure 5—figure supplement 2*). Despite the overall lower correlation between the immune cell phenotypes and the intact reservoir size, this tree nevertheless achieved 82.6% accuracy. The tree had six leaves (three with low intact reservoir, three with high intact reservoir) and relied on %Tn CD8 T cells, %CD107a−IFNγ+IL-2+TNFα− CD4 T cells, %CD127+CD4 T cells, %CD38−HLA-DR+Tcm CD8 T cells, and CD4 nadir. When we combined the 'high intact' leaves together and the 'low intact' leaves together, we observed a significant difference in median intact reservoir frequency between the groups (29.5/M for low intact and 101/M for high intact, Mann–Whitney $U$ test p-value is 9.59e−08, *Figure 5D*).

Finally, a third iteration of GOSDT generation was performed, classifying the cohort participants with respect to percentage intact of total proviruses (*Figure 5E*, *Figure 5—figure supplement 3*). This tree achieved 82.4% accuracy and relied on years of ART treatment, %KLRG1+ CD27 CD4 T cells, %CD127+ CD4 T cells, and recent clinical CD4 count. This tree has two leaves with low percent intact reservoir and three leaves with high percent intact reservoir. We observed a significant difference in the actual percent intact values between the model generated high percent intact and low percent intact leaves (median 5.7% for low percent intact leaves and median 15.3% for high percent intact leaves: Mann–Whitney $U$ test, p-value is 5.5e−10, *Figure 5F*).

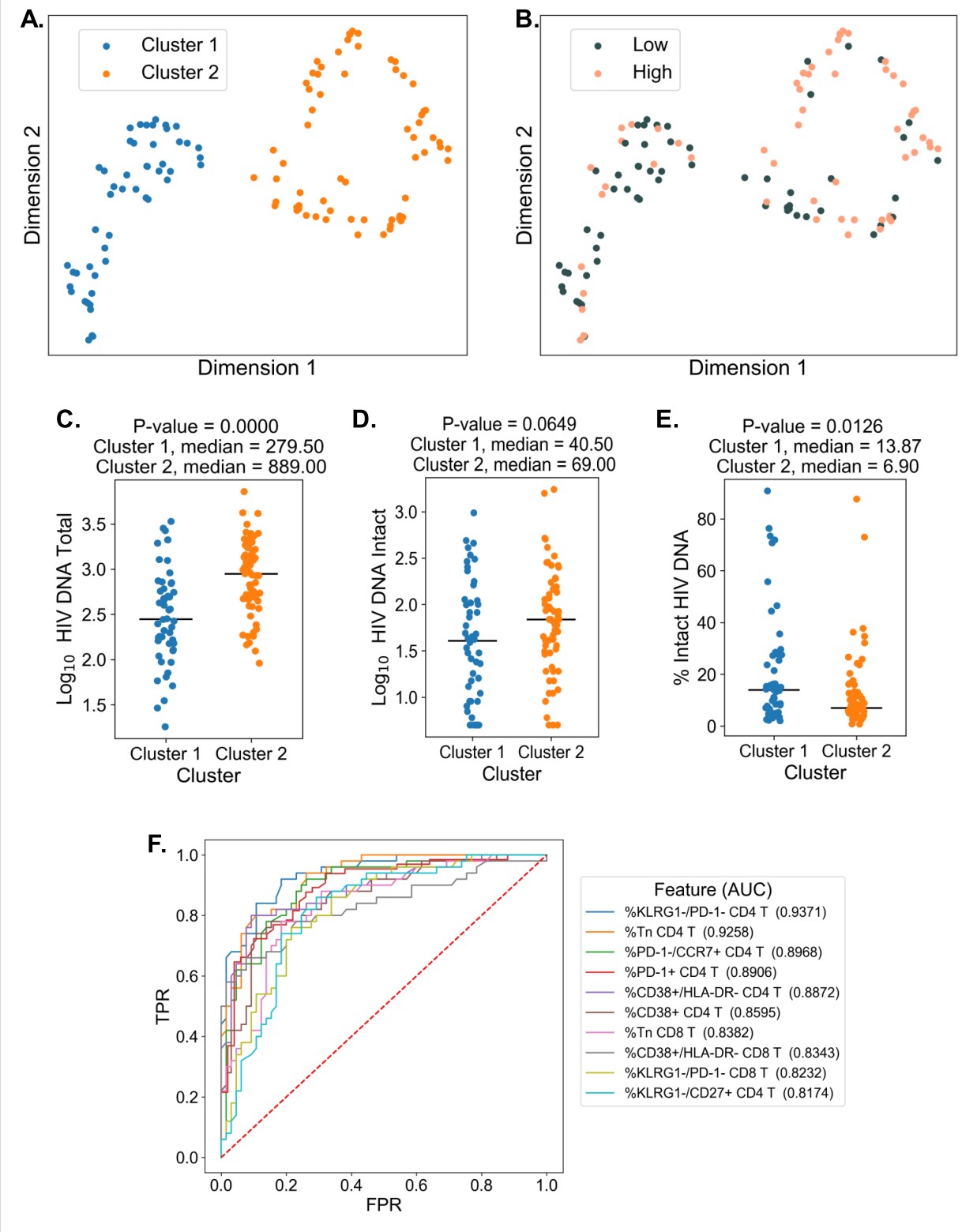

**Figure 4.** Dimension reduction reveals two major clusters of people with HIV (PWH) with distinct immune systems and reservoirs. (**A**) PaCMAP was applied to the data using the ten immune cell features with the highest area under the curve (AUC) values for classifying participants based on total reservoir frequency, and two clusters (clusters 1 and 2) are identified. (**B**) Same as A but data points are color-coded by total reservoir frequency (high = pink, low = gray). Total reservoir frequency (**C**), intact reservoir frequency (**D**), and percent intact (**E**) are shown for participants within each cluster. (**F**) Key immune cell features that distinguish cluster 1 from cluster 2 are identified by visualizing the features with the highest AUC values with respect

*Figure 4 continued on next page*

*Figure 4 continued*

to classifying cohort participants based on cluster membership. Axes represent the true positive rate (TPR) and the false positive rate (FPR) for each variable. Immune markers and clinical–demographic features are shown for each cluster in *Figure 4—figure supplement 1*.

The online version of this article includes the following figure supplement(s) for figure 4:

**Figure supplement 1.** Additional dimension reduction results.

**Figure supplement 2.** Principal component analysis (PCA) visualization.

Collectively, these decision trees highlight combinations of immune parameters that can accurately describe qualitatively high versus low total, intact, and percentage intact HIV DNA in a cohort of *n* = 115 PWH. This visualization serves as a basis for mechanistic hypotheses about the interactions of the immune system and HIV reservoir during long-term ART.

## ML algorithms identify immune feature combinations that predict high versus low reservoir metrics with ~70% test accuracy

Although clustering and decision tree analysis permit visualization and understanding of global structures within a dataset, we were curious if combinations of immune and clinical–demographic parameters could actually accurately predict, rather than only visualize, the size of the HIV reservoir. We considered five ML algorithms including Logistic Regression with L2 regularization (LR), CART, Support Vector Machines with RBF kernel (SVM), Random Forest (RF), and Gradient Boosted Trees (GBT). Initially, we attempted to predict reservoir frequency as a continuous variable, but found that models trained for this purpose performed poorly and tended to overfit to training sets (*Figure 6—figure supplement 1*). Thus, we focused our efforts on predicting high (above median) versus low (below median) reservoir frequency. For reservoir characteristics, we measured accuracy of the models over 10 random splits of our data into training and test sets (*Figure 6A* and *Supplementary file 1k*). Overall, we found that for total reservoir frequency, LR achieved highest mean classification accuracy (69.31%) in test data. The accuracy of these models is likely limited by the sample size (*n* = 115) and noise in the data, as evidenced by lower $R^2$ score of ML models for non-binarized HIV characteristics (*Figure 6—figure supplement 1*).

To examine the contribution of individual immune features to model performance, we examined LR coefficients for each immune cell variable in the model. Since LR coefficients are associated with expected change in log odds (based on $\log_e$), we can think about the coefficient $\beta$ for variable $X$ in the following way: increasing variable $X$ by one unit multiplies the odds of high reservoir frequency (probability that the reservoir size is high divided by the probability that the reservoir frequency is low) by $e^\beta$. In *Figure 6B*, we visualize the LR model for one data split among ten we considered for total reservoir frequency. For this split we observe that higher values of %NKG2A+CD4 T, %PD-1+Tn CD4 T, and %Tcm CD8 T are associated with a increased probability of total reservoir frequency being high. On the other hand, an increase in %Tn CD4 T decreases the odds of high reservoir frequency. The model visualized in *Figure 6B* achieved 75.86% training and 75% test accuracy.

Similarly, LR models performed best for predicting intact reservoir frequency compared to other methods (average 65.17% test accuracy, see *Figure 6C*). In *Figure 6D*, we visualize the LR model for one fixed data split. In this model we observed that higher values of %CD107a−IFNγ+IL-2+TNFα CD4 T and %CD127+ CD4 T, and %CD4 T are associated with lower probability of intact reservoir frequency being high, while higher values of %KLRG1+CD27+ CD8 T are associated with increased probability of intact reservoir frequency being high.

For analysis of percentage intact HIV DNA, we display training and test accuracy values in *Figure 6E*. For the visualized model for one data split (*Figure 6F*), an increase in %CD127+ CD4 T, years of ART, %KLRG1+PD-1+CD4 T, and %Tem CD4 T leads to a lower probability of percent intact HIV DNA being high.

Overall, these analyses demonstrate that we can use ML tools to construct models that can predict with approximately 70% accuracy whether a given PWH has qualitatively low or high total HIV DNA frequency, intact HIV DNA frequency or percentage of HIV DNA. Further studies with larger cohorts will likely improve the accuracy of these models.

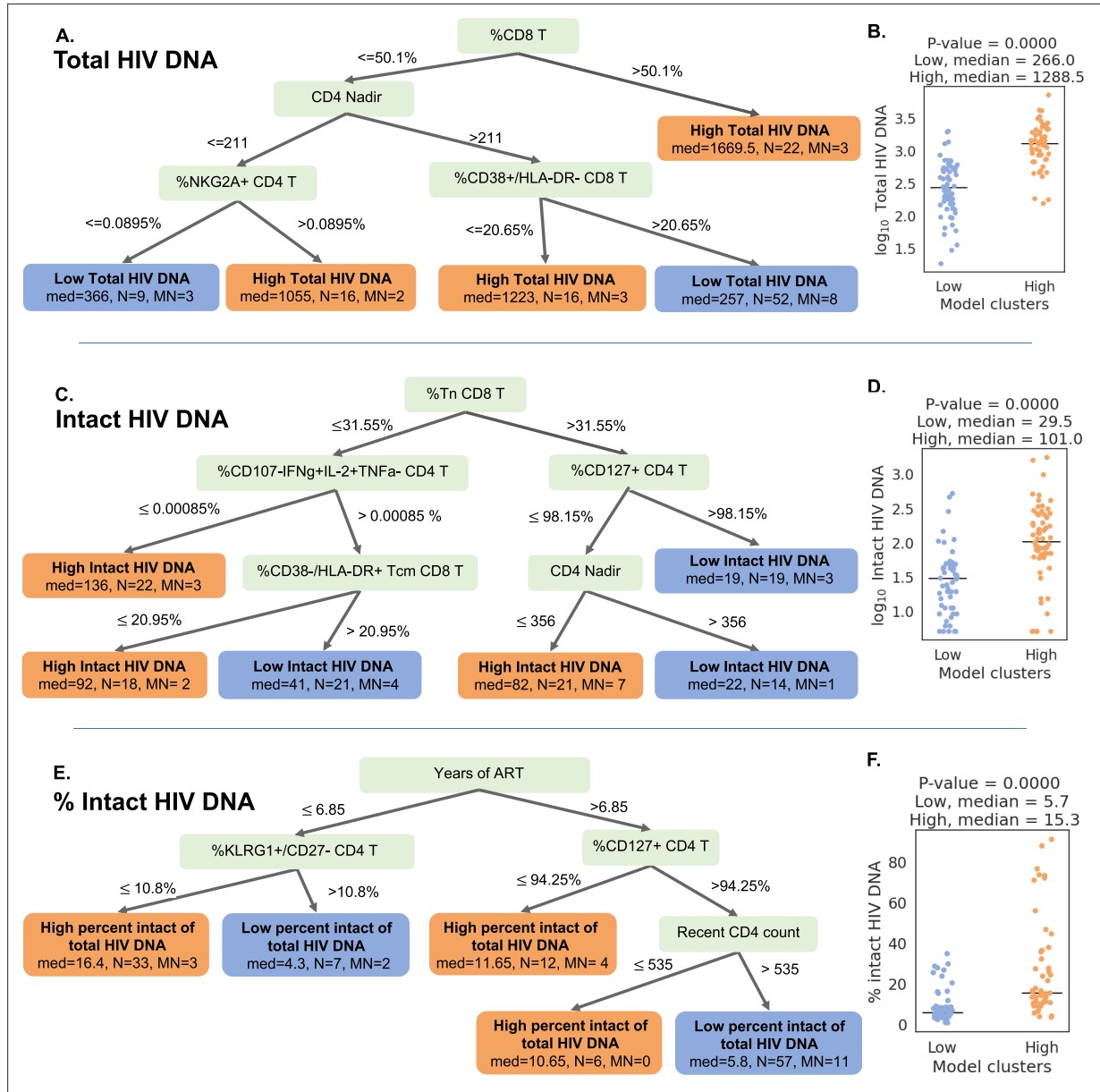

**Figure 5.** Decision tree visualization of the association of immune cell subsets with reservoir characteristics. (**A, C, E**) Host variables (immune cell frequencies, demographic, and clinical information) were used to visualize the people with HIV (PWH) dataset using the optimal sparse decision trees algorithm Generalized and Scalable Optimal Sparse Decision Trees (GOSDT). The overall set of PWH was classified as likely having high (above median, orange 'leaves') or low (below median, blue 'leaves') total reservoir frequency (**A**), intact reservoir frequency (**C**), and percent intact (**E**). In each leaf, 'med' denotes the median HIV characteristic of PWH, *N* is the number of PWH in the leaf, and MN is the number of mislabeled PWH. (**B, D, F**) PWH in model leaves associated with high (orange) or low (blue) reservoir frequency characteristics were aggregated and a Mann–Whitney *U* test was performed to determine statistical significance between the actual total reservoir frequency of the 'high' and 'low' groups for total reservoir frequency (**B**), intact reservoir frequency (**D**), and percent intact (**F**). For the percent intact tree we exclude participants with missing values of years of antiretroviral therapy (ART). For total and intact reservoir frequency, missing values of years of ART were imputed, however, since the trees do not use this variable, imputations do not influence results. Visualization trees are explained with sets of rules in figure supplements.

The online version of this article includes the following figure supplement(s) for figure 5:

**Figure supplement 1.** The total reservoir frequency visualization tree is explained with a set of rules.

**Figure supplement 2.** The intact reservoir frequency visualization tree is explained with a set of rules.

**Figure supplement 3.** The percent intact visualization tree is explained with a set of rules.

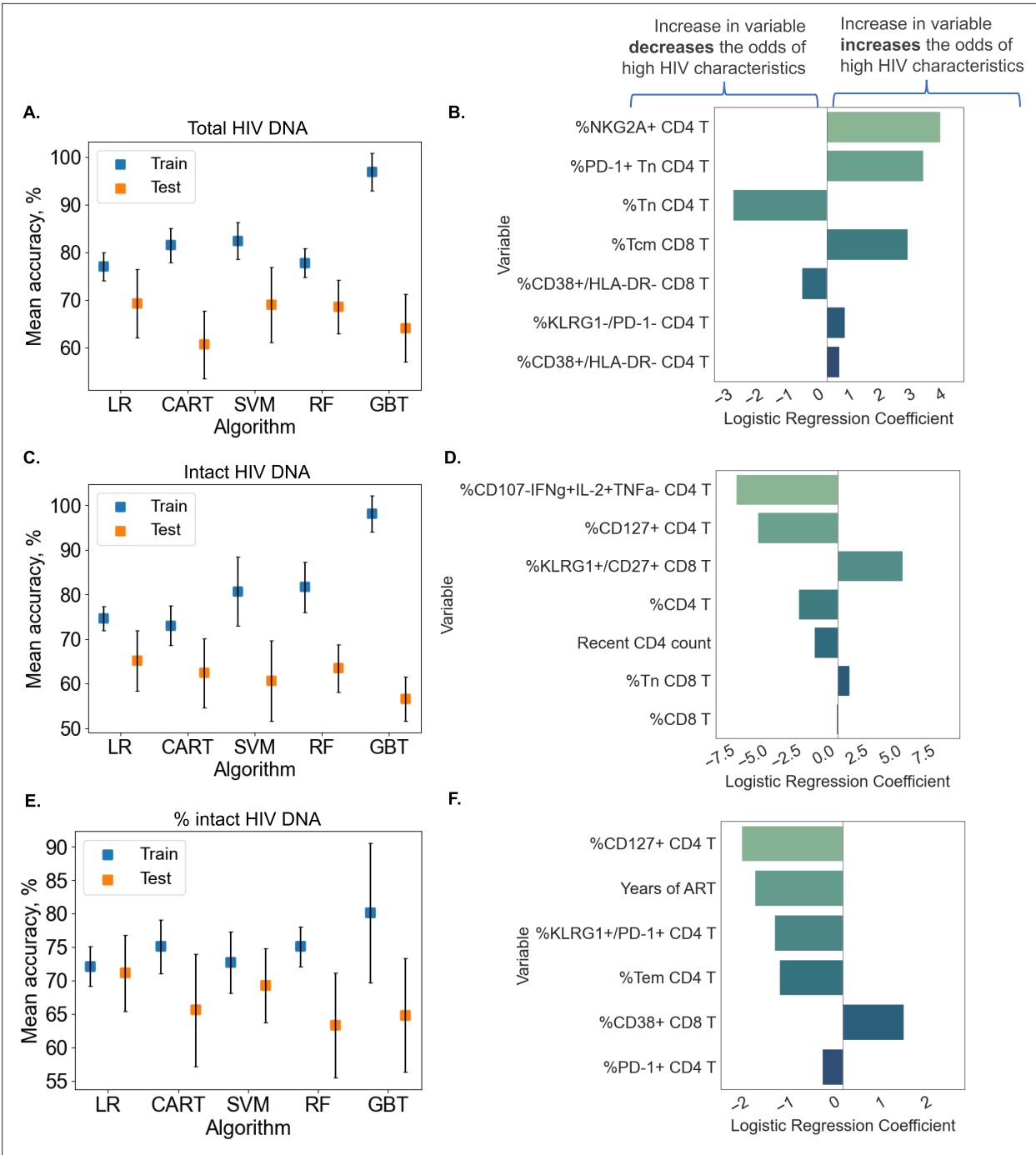

**Figure 6.** Predicting HIV reservoir characteristics with machine learning. Average training and test accuracies over 10 training and test data splits for Random Forest (RF), Gradient Boosted Trees (GBT), Support Vector Machines with RBF kernel (SVM), Logistic Regression (LR), and CART models for total reservoir frequency (**A**), intact reservoir frequency (**C**), and percent intact (**E**) are shown. For one split of training and test sets, LR models are visualized for total reservoir (**B**), intact reservoir (**D**), and percent intact (**F**). On the *y*-axis, we show variables used by the model, while the *x*-axis displays coefficient values for individual variables used by models. For percent intact models, we exclude participants with missing values of years of antiretroviral therapy (ART). For total and intact reservoir frequency, missing values of years of ART were imputed. The missing value of the CD4 nadir for one participant was imputed using the Multivariate Imputation by Chained Equations (MICE) algorithm.

The online version of this article includes the following figure supplement(s) for figure 6:

**Figure supplement 1.** Using machine learning to predict reservoir frequency.

## Discussion

In this cross-sectional study of peripheral blood samples from a well-characterized clinical cohort of 115 PWH on long-term ART, we used ML analysis of IPDA and high-parameter flow cytometry to better understand the associations between the HIV reservoir and a broad range of immunophenotypes. Using these approaches, specific HIV reservoir features (total HIV DNA frequency, intact HIV DNA frequency, percentage intact HIV DNA) and clinical–demographic variables (age, sex, CD4 nadir, recent CD4 count, years on ART, years of HIV infection prior to initiating ART) were identified that predict specific immunophenotypes in PWH on long-term ART. Distinct host immune correlates of intact versus total HIV DNA frequency were also identified. Further, combinations of immune parameters that classified and predicted whether a given PWH has high or low total, intact, or percentage intact HIV DNA were defined. Collectively, this study confirms existing knowledge about HIV reservoir biology and identifies novel associations for further investigation. Additionally, from a methodological perspective, this work also demonstrates the utility of specific data science and ML approaches for HIV reservoir studies. Development and application of these analytic approaches are important to enable biological insights from high-parameter studies of the HIV reservoir and other complex biology, particularly with the advent of high-dimensional techniques such as CYTOF and scRNAseq.

Previous studies have sought to identify immune and other correlates of the total HIV DNA reservoir (*Horsburgh et al., 2020*; *Chomont et al., 2009*; *Cockerham et al., 2014*; *Hatano et al., 2013*; *Ruggiero et al., 2015*; *Lee et al., 2019*; *Huang et al., 2023*; *Banga et al., 2016*; *Fromentin et al., 2019*; *Fromentin et al., 2016*; *Pardons et al., 2019*; *Dufour et al., 2023*; *Zhu et al., 2023*; *Wu et al., 2023*; *Gálvez et al., 2021*; *Bernal et al., 2023*; *Astorga-Gamaza et al., 2023*; *Fisher et al., 2023*; *Dubé et al., 2023*; *Takata et al., 2023*). These studies have identified several activation/exhaustion markers that correlate with total HIV DNA, including the CD4/CD8 ratio, T cell expression of HLA-DR, CD38, PD-1, CTLA-4, and LAG-3, and NK cell KLRG1 expression. However, not all studies have found associations of total HIV DNA and immune subsets, particularly for CD4 T cell HLA-DR and CD38 expression (*Poizot-Martin et al., 2013*; *Gandhi et al., 2017*; *Gálvez et al., 2021*; *Besson et al., 2014*) during long-term ART. There is also growing appreciation for additional factors that correlate with the HIV reservoir. Specifically, the size of the reservoir has been linked to the abundance of specific bacteria species in the intestinal microbiome (*Borgognone et al., 2022*). Also, genetic studies have indicated the existence of single-nucleotide polymorphisms (SNPs) that are associated with reservoir size, including SNPs in MX1 and IRF7 (*Siegel et al., 2023*; *Thorball et al., 2020*; *Zhang et al., 2021*).

The correlative results from this present study corroborate many of these studies, and provide additional insights. A number of correlations between total HIV DNA and immune activation and exhaustion were identified. There were weak to moderate positive correlations observed between total HIV DNA frequency and CD4 T cell expression of NKG2A, PD-1, KLRG1, and HLA-DR as well as CD8 T cell expression of PD-1 and CCR7. Total HIV DNA frequency was inversely correlated with CD8 T cell expression of CD38, CD4 T cell expression of CD27, the frequency of naive CD4 and CD8 T cells, and the CD4/CD8 ratio. Overall our results support the hypothesis that larger HIV reservoir size is associated with increased frequencies of cells with an activated or exhausted phenotype, with depletion of naive cells and with an increased proportion of central memory CD8 T cells. The correlation of NKG2A expression in CD4 T cells with the total reservoir size represents a novel observation that merits further investigation. Only a very small percentage of CD4 T cells express NKG2A (less at 1%), and the biological significance for the correlation of these cells with the HIV reservoir is unknown. NGK2A is known to function as an inhibitory receptor for NK cells and for CD8 T cells, but its role in CD4 T cells is less clear (*Creelan and Antonia, 2019*).

When we examined immune correlates of intact HIV DNA measured by the IPDA, many of the same relationships as described above were noted, but with decreased magnitude and often lacking statistical significance. This is consistent with a recent study using the IPDA that did not find associations of intact HIV DNA with CD4 T cell HLA-DR and CD38 co-expression (*Gandhi et al., 2021*), as well as a study using near-full-length viral sequencing in single cells that found only slight enrichment of HLA-DR or PD-1 in cells with genetically intact, inducible proviruses (*Dufour et al., 2023*). The stronger relationship between the immune cell phenotypes and the total reservoir, rather than the intact reservoir, should be investigated further. We speculate that this association could be driven by the more numerous defective proviruses promoting inflammation or immune activation through innate immune sensing pathways, or a greater level of transcriptional activity for defective proviruses. This is certainly

plausible giving the growing appreciation for the immunogenicity and high frequency of defective proviruses within the HIV reservoir (*Tumpach et al., 2023*; *Singh et al., 2023*; *Ishizaka et al., 2016*; *Martin et al., 2022*). It is important to note, however, that it is also possible that the stronger association of total proviral burden with immunophenotype reflects persistent damage caused to the immune system before therapy, and that this association between the total reservoir and immune cells may not result from ongoing interactions between HIV and the immune system during therapy.

Uniquely, CD127 expression on CD4 T cells was significantly associated with the intact reservoir frequency but not the total reservoir. Specifically, the frequency of CD127− CD4 T cells was positively correlated with the frequency of intact proviruses. CD127 is the alpha-chain IL-7 receptor and is expressed on long-lived memory T cells that are depleted during untreated HIV infection. Post-ART, the frequency of CD127+ memory CD4 subsets in PWH increases over time, albeit slowly (*Sponaugle et al., 2023*). Interestingly, CD127+ cells are enriched in latent HIV infection in tissues (*Hsiao et al., 2020*), and IL-7 is associated with slower natural reservoir decay (*Chomont et al., 2009*). Furthermore, IL-7 signaling is known to drive expression of VLA-4, which was recently reported to highly enrich for genetically intact, inducible proviruses (*Dufour et al., 2023*). This observational finding that intact but not total HIV DNA inversely correlates with CD127 expression on CD4 T cells requires further investigation.

DR ML approaches identified two robust clusters of PWH when using total HIV DNA reservoir-associated immune cell frequencies. These clusters exhibited distinct reservoir characteristics, with one cluster being enriched with participants with a larger total reservoir size, a lower percentage of intact viruses, a lower frequency of naive T cells and elevated expression of activation and exhaustion markers, while the other contained PWH with smaller total reservoirs, higher percentage of intact viruses, higher frequency of naive T cells, and lower expression of activation and exhaustion markers. The cluster membership was notably associated with age and time on therapy. The existence of two distinct clusters of PWH with different immune features and reservoir characteristics could have implications for HIV cure strategies. These two clusters are also interesting in the context of a previous study examining reservoir characteristics which proposed two major types of viral reservoir within PWH: one type with smaller reservoirs that were enriched in Tcm cells and associated with lower levels of Ki67+ cells and immune activation, and another type with larger reservoirs that were enriched in Ttm cells and associated with higher levels of Ki67+ cells and immune activation (*Chomont et al., 2009*). It is possible that these two proposed types of reservoirs represent the two clusters we observe using immune markers and DR, although we were unable to examine the frequency of HIV proviruses specifically within Tcm and Ttm cells in this study.

In addition to unsupervised DR approaches, we were able to construct simple, interpretable decision trees that describe our cohort of PWH with more than 80% accuracy with respect to the size of the reservoir. Furthermore, when we constructed decision trees to classify PWH separately based on total reservoir frequency, intact reservoir frequency and percentage intact, distinct sets of features were important (though some were the same) for each of these models. The decision tree analyses highlight complex immune parameter combinations that serve as a basis for unique hypotheses about HIV reservoir biology. One striking example of ML approaches identifying potential reservoir biology is the percentage intact HIV DNA decision tree. As expected, years of ART was critical to predict the percentage of intact proviruses, but CD4 T cell CD127 expression, CD4 T cell KLRG1 expression, and recent CD4 count also played a significant role.

LR models that we trained to classify high versus low total reservoir, intact reservoir, and percentage intact were able to predict out-of-sample with ~70% accuracy on average. Each of these models relied on five features only and were simple enough for us to visualize and assess how important each variable is for every characteristic of reservoir size. These models corroborated the potential biological relationships identified in the decision tree analysis, and confirmed that these variables could actually be used to predict out-of-sample whether total, intact, or percentage intact HIV DNA was qualitatively high or low. The success of our predictive models was likely limited by the relative small size of the datasets used to train and test them (*n* = 115). As such, larger datasets will be required to generate more accurate models that can be used for reliable prediction of reservoir characteristics. Additionally, it is likely that unmeasured factors and biological noise also contribute to reservoir size.

Our findings should be considered in the light of some inherent limitations and caveats. This study is cross-sectional in nature and is primarily observational, so the findings should be interpreted with

caution. As with any observational study, confounding bias may influence our results, particularly for correlative analyses. The associations between the HIV reservoir and the immune system we observed were fairly weak, despite being statistically significant. Similarly, the predictive ability of our ML models was modest (70%) for classifying PWH as having an above or below median reservoir. As mentioned above, our cohort is likely too small to generate accurate predictive ML models, and consists of mostly male participants, which could bias the results. We also cannot rule out the influence of additional unmeasured confounding factors. In particular, the length of untreated infection prior to ART could be a major contributor to the immune signatures we observe. In this study, the participants had all been treating during chronic infection, and different associations might be observed in PWH who are treated during acute infection. Another limitation to this study is that our analyses only considered cells and proviruses in the peripheral blood, and this may not represent HIV reservoirs and immune cells located in tissues known to harbor latent HIV such as gut, spleen, and brain. Our analysis of surface protein phenotypes also does not detect potentially important associations between the reservoir and transcriptional pathways within these cell types. Future work using single-cell RNAseq from a similar cohort may help to reveal deeper layers of associations that are invisible to the current analysis. As is always the case with correlation studies, its is important to be aware that these associations are only correlations and do not necessarily represent functional or mechanistic relationships. Even so, these correlations are still useful as a source of novel hypotheses that could be tested in subsequent studies.

Nevertheless, our study does have some strengths compared to previous work in this area, specifically a highly detailed flow panel, allowing us to quantify numerous immune subsets with a high degree of resolution. Also, the use of the IPDA allows us to separately examine the association of the immune system with intact and total viral DNA. It is however, important to note that IPDA does not measure all proviral sequences in PWH and is likely failing to identify some sequences due to proviral polymorphisms or dual deletions in the two IPDA amplicons. Also, a minor fraction of IPDA+ proviruses are actually defective due to the limited specificity of using two short amplicons to select for full-length intact proviruses (*Bruner et al., 2019*; *Gaebler et al., 2021*).

Overall, these findings suggest a complex concert of immune recovery, HIV reservoir dynamics, and intrinsic host factors (age, biological sex) that shapes host immunophenotype, even after years of ART. Mechanistic work will be needed to fully dissect the dynamic relationship between the immune system and the reservoir. Nevertheless, these findings support a model in which ongoing interaction between the HIV reservoir and the host immune cells continue to drive an association, albeit a relatively minor one, with persistent CD4 T cell activation during long-term ART. Notably, both intact and defective proviruses appear to contribute to this immune signature of HIV latency during long-term ART. This study identifies several areas for future investigation. The question of whether the HIV reservoir directly drives persistent immune activation could be addressed directly by specifically suppressing viral transcription during therapy and examining the impact on the immune system. The stronger association of the intact reservoir with CD127 expression in CD4 T cells is an intriguing observation that warrants further validation and explanation. This observation could, for example, be explained by a model in which the intact reservoir is preferentially located in cells with a distinct pattern of CD127 expression. These ideas, along with other future studies exploring the interplay of the host immune system and HIV reservoir might leverage new high-dimensional technologies such as scRNAseq and mass cytometry. The ML approaches described and applied in this study may be particularly useful for gaining biological insights into these high-parameter datasets. In particular, simple and interpretable ML tools for visualizing the data, such as GOSDT, will help to identify specific limited sets of immune parameters that associate with reservoir size and which could represent targets for therapies designed to reduce reservoir persistence.

## Methods
### Cohort and sample collection

115 PWH were recruited from two clinical sites. 66 PWH were recruited at Duke University medical center and 49 PWH at UNC Chapel Hill. Peripheral blood was collected and PBMCs were obtained by Ficoll separation, then frozen in 90% fetal bovine serum 10% dimethylsulfoxide (*Falcinelli et al., 2023*; *Falcinelli et al., 2020*; *Gay et al., 2022*; *Falcinelli et al., 2021*).

## Flow cytometry

Cell preparation and staining followed previously described methods (*Healy and Murdoch, 2016*). All antibodies were titrated to optimize signal-to-noise ratio on PBMCs prior to use, assuming a 50-µl staining volume. Cell viability was examined using Zombie-NIR Fixable Viability Dye (0.4 µl per 50 µl staining volume; Biolegend). Antibodies used for surface staining were as follows: KLRG1-BV421 (SA231A2 clone, Biolegend, CD45RA-PacBlue (H100 clone, BL), CD8-BV570 (RPA-T8 clone, Biolegend), CD127-BV605 (A019D5 clone, Biolegend), CD56-BV650 (5.1H11 clone, Biolegend), CCR7-BV711 (G043H7 clone, Biolegend), CD27-BV750 (O323 clone, Biolegend), PD1-VioBright515 (REA165 clone, Miltenyi), NKG2A-PE-Vio615 (REA110 clone, Miltenyi), CD16-PerCP-Cy5.5 (33G8 clone, Biolegend), CD38-PCPeF710HB7 clone, TF), CD14-SparkNIR685 (63D3 clone, Biolegend), CD19-SparkNIR685 (HIB19 clone, Biolegend), and HLA-DR-APC-F750 (L243 clone, Biolegend). Antibodies used for intra-cellular staining were as follows: CD3-BV480 (UCHT1 clone, BD), CD4-PerCP (L200 clone, BD), IFN-g-PE-Cy7 (4S.B3 clone, Biolegend), IL-2-APC (MQ1-17H12 clone, Biolegend), and TNFα-AF700 (Mab11 clone, Biolegend). For HIV peptide stimulation, the cells were incubated with peptide mixes from Gag, Pol and Env proteins at 0.2 µg/ml in the presence of brefeldin A and monesin for 6 hr. Samples were analyzed using a Cytek Aurora spectral flow cytometer. Flow cytometry analysis was performed in FlowJo v10.8 software.

## Intact proviral DNA assay

Cryopreserved samples of PBMCs from each study participant were viably thawed. A portion was used for immunophenotyping as described above, and the remainder were subjected to total CD4 T cell negative selection with the StemCell Technologies EasySep Human CD4+ T Cell Enrichment Kit (Cat#19052). CD4 T cell DNA was extracted using the QIAamp DNA Mini Kit and quantified on a NanoDrop 1000 (Thermo Fisher Scientific). IPDA was performed as originally described (*Bruner et al., 2019*), with a validated PCR annealing temperature modification to increase signal-to-noise ratio (*Falcinelli et al., 2021*). Gating for positive droplets was set using negative (DNA elution buffer and HIV-seronegative CD4 T cell DNA), and positive (Integrated DNA Technologies gblock amplicon) control wells processed in parallel. DNA shearing index values were similar to those reported previously (median, 0.33; IQR, 0.31–0.34) (*Falcinelli et al., 2020*; *Gay et al., 2022*; *Falcinelli et al., 2021*). A median of $1.04 \times 10^6 (Q1 - 8.71 \times 10^5, Q3 - 1.16 \times 10^6)$ cell equivalents were assessed for each donor.

## Statistics

### Variable data analysis

To create binarized labels that represent reservoir characteristics, we split the data at the median, which is 553/M for total reservoir frequency, 53/M for intact reservoir frequency, and 8.64% for percent intact. For every variable (immune cell frequency, demographics, and clinical information), we computed Spearman correlation and an AUC value. To compute the AUC value, we first created an ROC curve by plotting the true positive rate versus the true negative rate for every cell subset frequency. AUC is then computed as the area under the ROC curve using the trapezoidal rule.

To assess the importance of clinical, demographic information, and HIV reservoir characteristics in describing the immune markers, we performed an LOCO analysis. We trained a linear regression model ($M_1$) with intercept based on all variables. We removed each variable one at a time and trained a new model without this variable. Then we computed the difference in $R^2$ score before and after dropping the variable. Variance inflation factors (VIFs) for age, sex, years of ART, CD4 nadir, recent CD4 account, and years of HIV infection prior to HIV infection were less than 2, indicating an acceptable level of correlation between these independent variables (*James et al., 2013*). However, higher VIF values were observed for intact and total HIV DNA, therefore separate LOCO inference analyses were conducted in order to avoid artificial fluctuations in model fit due to multicollinearity.

To examine the connection between CD4/CD8 and (CD127+ CD4)/CD8 ratio and HIV characteristics (*Figure 1—figure supplement 1*), we first computed the logarithm ($\log_{10}$) of the intact, total and percent intact as the outcome and normalized outcome of each immune feature (meaning that our values of ratio and outcome are between 0 and 1). We removed outliers using the DBSCAN algorithm (*Ester et al., 1996*), which is an unsupervised clustering algorithm that groups data points based on density into a single cluster. For DBSCAN, we set the parameter that determines radius of a circle

around each point (that is used to compute density) as 0.15 and minimum number of samples in each cluster as 10. Then we fitted the remaining data points using Linear Regression.

### Data visualization

We used PaCMAP (*Wang et al., 2021*) to reduce the dimensionality of the dataset to a two-dimensional space. Due to the relatively small sample size of our dataset ($n$ = 115), we set the number of neighbors to 5. Additionally, we used GOSDT (*Lin et al., 2020*) which computes sparse optimal trees to identify patterns in the dataset. For all the visualization trees, we set the depth budget parameter to 4 and the regularization parameter to 0.02.

### Training and generalization

Our training procedure for classification is described in *Supplementary file 1j*, which is a standard ML pipeline with an added search for the number of variables. For the full hyperparameters list see *Supplementary file 1k*. We normalized continuous variables to have values between 0 and 1. For logistics regression models, we set the regularization parameter of $\ell-2$ regularization based on cross-validation. The values are 10 for total reservoir frequency and percent intact and 100 for intact reservoir frequency.

To predict reservoir frequency based on immune cell frequencies, clinical and demographic information (*Figure 6—figure supplement 1*), similar to classification, we sought to find the set of model parameters and types of models that would be able to fit the dataset without overfitting. We used several approaches: LR, Ridge Regression, Kernel Regression with RBF kernel, Decision Tree Regressor, RF, and GBT. We followed the procedure described in *Supplementary file 1j*, optimized the $R^2$ score, returned the mean and standard deviation of the $R^2$ score of training and test sets, and chose features based on the absolute value of the correlation coefficient. We normalized variables to have values between 0 and 1. We then fitted the models to the normalized natural logarithm ($\log_e$) of total, intact reservoir frequency, and percent intact and then further normalization makes outcomes to be between 0 and 1. For total and intact reservoir frequency, we visualized the Ridge Regression model, which performed the best based on mean test $R^2$ score (*Figure 6—figure supplement 1D, E, G, H*). We set the regularization parameter to 1, as this value was best on average according to cross-validation for total and intact reservoir frequency. For percent intact, we visualized Linear Regression based on four variables (*Figure 6—figure supplement 1F, I*) as it achieved the second highest $R^2$ score after Kernel Ridge based on 25 variables.

### Study approval

Written informed consent was obtained for all study participants. The study design was reviewed and approved by IRB for both Duke University and UNC Chapel Hill (Study numbers: 21-0117 and 08-1575).

## Acknowledgements

This work was supported by the following grants from the National Institutes of Health: NIAID R01 AI143381 (EPB), NIAID UM1 AI164567 (DM Margolis), NIDA R61 DA047023 (EPB), NIDA R01 DA054994 (CDR), and NIAID F30 AI145588 (SDF).

## Additional information

### Funding

| Funder | Grant reference number | Author |
| --- | --- | --- |
| National Institute of Allergy and Infectious Diseases | R01 AI143381 | Edward P Browne |
| National Institute on Drug Abuse | R61 DA047023 | Edward P Browne |

| Funder | Grant reference number | Author |
|---|---|---|
| National Institute on Drug Abuse | R01 DA054994 | Cynthia D Rudin |
| National Institute of Allergy and Infectious Diseases | UM1 AI164567 | David M Margolis |
| National Institute of Allergy and Infectious Diseases | F30 AI145588 | Shane Falcinelli |

The funders had no role in study design, data collection, and interpretation, or the decision to submit the work for publication.

## Author contributions

Lesia Semenova, Software, Formal analysis, Investigation, Visualization, Writing – original draft, Writing – review and editing; Yingfan Wang, Investigation, Visualization; Shane Falcinelli, Investigation, Writing – original draft, Writing – review and editing; Nancie Archin, Data curation, Investigation, Writing – original draft, Writing – review and editing; Alicia D Cooper-Volkheimer, Data curation, Investigation; David M Margolis, Resources, Funding acquisition, Writing – original draft, Writing – review and editing; Nilu Goonetilleke, Writing – original draft, Writing – review and editing; David M Murdoch, Resources, Supervision, Funding acquisition, Investigation, Writing – original draft, Project administration, Writing – review and editing; Cynthia D Rudin, Data curation, Software, Formal analysis, Supervision, Funding acquisition, Writing – original draft, Writing – review and editing; Edward P Browne, Conceptualization, Supervision, Funding acquisition, Investigation, Writing – original draft, Project administration, Writing – review and editing

## Author ORCIDs

Nilu Goonetilleke (iD) https://orcid.org/0000-0003-2278-1656
David M Murdoch (iD) https://orcid.org/0000-0001-7201-7950
Edward P Browne (iD) https://orcid.org/0000-0001-9070-7015

## Ethics

Written informed consent was obtained for all study participants. The study design was reviewed and approved by IRB for both Duke University and UNC Chapel Hill.

Reviewer #1 (Public review) https://doi.org/10.7554/eLife.94899.3.sa1
Author response https://doi.org/10.7554/eLife.94899.3.sa2

# Additional files

## Supplementary files
• MDAR checklist

• Supplementary file 1. Additional tables with raw data values for main figures. (**a**) Intact, total reservoir frequency, and %intact for demographic subgroups. (**b**) Immune subsets characteristics. (**c**) Host features correlate with HIV reservoir characteristics. (**d**) People with HIV (PWH) immune features correlate with years of antiretroviral therapy (ART). (**e**) Multicolinearity analysis for variables used in models to predict immunophenotypes. (**f**) Adjusted $R^2$ scores and differences in adjusted $R^2$ for leave-one-covariate-out (LOCO) analysis for the model that contains total reservoir frequency. (**g**) Adjusted $R^2$ scores and differences in adjusted $R^2$ for LOCO analysis for the model that contains intact reservoir frequency. (**h**) Adjusted $R^2$ scores and differences in adjusted $R^2$ for LOCO analysis for the model that contains percent intact. (**i**) Host features classify PWH with respect to HIV reservoir characteristics. (**j**) Training procedure for classification (regression). (**k**) Ranges of hyperparameters values that we used to perform grid search for classification and regression.

## Data availability

Underlying data available at: https://doi.org/10.15139/S3/3CFVIU. Code is available at: https://github.com/lesiasemenova/ML_HIV_reservoir (copy archived at *Semenova, 2024*).

The following dataset was generated:

| Author(s) | Year | Dataset title | Dataset URL | Database and Identifier |
|---|---|---|---|---|
| Browne EP | 2024 | Machine learning approaches identify immunologic signatures of total and intact HIV DNA during long-term antiretroviral therapy | https://doi.org/10.15139/S3/3CFVIU | UNC Dataverse, 10.15139/S3/3CFVIU |

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
