## [Editor Report · eLife assessment]

Semenova et al. have studied a large cross-sectional cohort of people living with HIV on suppressive antiretroviral therapy and performed high dimensional flow-cytometry for analysis with data science/machine learning approaches to investigate associations of immunological and clinical parameters and intact/total HIV DNA levels (and categorizations). The study is **useful** in introducing these new methods and large data set and appears mostly **solid**, though some of the claims were **incompletely** supported by the modeling results. The authors have revised the text to fairly reflect their results, yet open questions remain about utility, particularly as to the value of categorical classification (vs continuous measurement) of reservoir size.

---

## [Referee Report · Reviewer #1 (Public review)]

On responding to the first round of reviews, the authors have nicely adjusted their wording and fairly describe the results of their study. Certain markers were identified for further investigation. Yet, an overall non-obvious relationship between immune markers and HIV reservoirs has been shown previously, and despite the attempt to leverage powerful ML algorithms, they are not magical and cannot reveal strong relationships that fundamentally do not exist. In addition, categorical classification is for now hard to interpret and the more powerful ML algorithms do not seem to outperform more classic regression methods. Therefore, it remains relatively hard to evaluate the utility of this kind of study.

Initial summary:

Semenova et al. have studied a large cross-sectional cohort of people living with HIV on suppressive ART, N=115, and performed high dimensional flow-cytometry to then search for associations between immunological and clinical parameters and intact/total HIV DNA levels.

A number of interesting data science/ML approaches were explored on the data and the project seems a serious undertaking. However, like many other studies that have looked for these kinds of associations, there was not a very strong signal. Of course the goal of unsupervised learning is to find new hypotheses that aren't obvious to human eyes, but I felt in that context, there were (1) results slightly oversold, (2) some questions about methodology in terms mostly of reservoir levels, and (3) results were not sufficiently translated back into meaning in terms of clinical outcomes.

Strengths:

The study is evidently a large and impressive undertaking and combines many cutting edge statistical techniques with a comprehensive experimental cohort of people living with HIV, notably inclusive of populations underrepresented in HIV science. A number of intriguing hypotheses are put forward that could be explored further. Data will be shared and could be a useful repository for more specific analyses.

Weaknesses:

Despite the detailed experiments and methods, there was not a very strong signal for variable(s) predicting HIV reservoir size. The spearman coefficients are ~0.3, (somewhat weak, and acknowledged as such) and predictive models reach 70-80% prediction levels, though of sometimes categorical variables that are challenging to interpret.

There are some questions about methodology, as well as some conclusions that are not completely supported by results, or at minimum not sufficiently contextualized in terms of clinical significance. Edit, authors have substantially revised the text.

On associations: the false discovery rate correction was set at 5%, but data appear underdetermined with fewer observations than variables (144vars > 115ppts), and it isn't always clear if/when variables are related (e.g inverses of one another, for instance %CD4 and %CD8).

The modeling of reservoir size was unusual, typically intact and defective HIV DNA are analyzed on a log10 scale (both for decays and predicting rebound). Also sometimes in this analysis levels are normalized (presumably to max/min?, e.g. S5), and given the large within-host variation of level we see in other works, it is not trivial to predict any downstream impact of normalization across population vs within person. Edit, fixed.

Also, the qualitative characterization of low/high reservoir is not standard, and naturally will split by early/later ART if done as above/below median. Given the continuous nature of these data it seems throughout that predicting above/below median is a little hard to translate into clinical meaning.

Lastly, work is comprehensive and appears solid, but the code was not shared to see how calculations were performed. Edit, fixed.

---

## [Author Response]

The following is the authors’ response to the original reviews.

**Public Reviews:**

**Reviewer #1 (Public Review):**
Summary:Semenova et al. have studied a large cross-sectional cohort of people living with HIV on suppressive ART, N=115, and performed high dimensional flow cytometry to then search for associations between immunological and clinical parameters and intact/total HIV DNA levels.A number of interesting data science/ML approaches were explored on the data and the project seems a serious undertaking. However, like many other studies that have looked for these kinds of associations, there was not a very strong signal. Of course, the goal of unsupervised learning is to find new hypotheses that aren't obvious to human eyes, but I felt in that context, there were (1) results slightly oversold, (2) some questions about methodology in terms mostly of reservoir levels, and (3) results were not sufficiently translated back into meaning in terms of clinical outcomes.

We appreciate the reviewer’s perspective. In our revised version of the manuscript, we have attempted to address these concerns by more adequately explaining the limitations of the study and by more thoroughly discussing the context of the findings. We are not able to associate the findings with specific clinical outcomes for individual study participants but we speculate about the overall biological meaning of these associations across the cohort. We cannot disagree with the reviewer, but we find the associations statistically significant, potentially reflecting real biological associations, and forming the basis for future hypothesis testing research.

Strengths:The study is evidently a large and impressive undertaking and combines many cutting-edge statistical techniques with a comprehensive experimental cohort of people living with HIV, notably inclusive of populations underrepresented in HIV science. A number of intriguing hypotheses are put forward that could be explored further. Sharing the data could create a useful repository for more specific analyses.

We thank the reviewer for this assessment.

Weaknesses:Despite the detailed experiments and methods, there was not a very strong signal for the variable(s) predicting HIV reservoir size. The Spearman coefficients are ~0.3, (somewhat weak, and acknowledged as such) and predictive models reach 70-80% prediction levels, though sometimes categorical variables are challenging to interpret.

We agree with the reviewer that individual parameters are only weakly correlated with the HIV reservoir, likely reflecting the complex and multi-factorial nature of reservoir/immune cell interactions. Nevertheless, these associations are statistically significant and form the basis for functional testing in viral persistence.

There are some questions about methodology, as well as some conclusions that are not completely supported by results, or at minimum not sufficiently contextualized in terms of clinical significance. On associations: the false discovery rate correction was set at 5%, but data appear underdetermined with fewer observations than variables (144vars > 115ppts), and it isn't always clear if/when variables are related (e.g inverses of one another, for instance, %CD4 and %CD8).

When deriving a list of cell populations whose frequency would be correlated with the reservoir, we focused on well-defined cell types for which functional validation exists in the literature to consider them as distinct cell types. For many of the populations, gating based on combinations of multiple markers leads to recovery of very few cells, and so we excluded some potential combinations from the analysis. We are also making our raw data available for others to examine and find associations not considered by our manuscript.

The modeling of reservoir size was unusual, typically intact and defective HIV DNA are analyzed on a log10 scale (both for decays and predicting rebound). Also, sometimes in this analysis levels are normalized (presumably to max/min?, e.g. S5), and given the large within-host variation of level we see in other works, it is not trivial to predict any downstream impact of normalization across population vs within-person.

We have repeated the analysis using log10 transformed data and the new figures are shown in Figure 1 and S2-S5.

Also, the qualitative characterization of low/high reservoir is not standard and naturally will split by early/later ART if done as above/below median. Given the continuous nature of these data, it seems throughout that predicting above/below median is a little hard to translate into clinical meaning.

Our ML models included time before ART as a variable in the analysis, and this was not found to be a significant driver of the reservoir size associations, except for the percentage of intact proviruses (see Figure 2C). Furthermore, we analyzed whether any of the reservoir correlated immune variables were associated with time on ART and found that, although some immune variables are associated with time on therapy, this was not the case for most of them (Table S4). We agree that it is challenging to translate above or below median into clinical meaning for this cohort, but we emphasize that this study is primarily a hypothesis generating approach requiring additional validation for the associations observed. We attempted to predict reservoir size as a continuous variable using the data and this approach was not successful (Figure S13). We believe that a significantly larger cohort will likely be required to generate a ML model that can accurately predict the reservoir as a continuous variable. We have added additional discussion of this to the manuscript.

Lastly, the work is comprehensive and appears solid, but the code was not shared to see how calculations were performed.

We now provide a link to the code used to perform the analyses in the manuscript, https://github.com/lesiasemenova/ML_HIV_reservoir.

**Reviewer #2 (Public Review):**
Summary:Semenova et. al., performed a cross-sectional analysis of host immunophenotypes (using flow cytometry) and the peripheral CD4+ T cell HIV reservoir size (using the Intact Proviral DNA Assay, IPDA) from 115 people with HIV (PWH) on ART. The study mostly highlights the machine learning methods applied to these host and viral reservoir datasets but fails to interpret these complex analyses into (clinically, biologically) interpretable findings. For these reasons, the direct translational take-home message from this work is lost amidst a large list of findings (shown as clusters of associated markers) and sentences such as "this study highlights the utility of machine learning approaches to identify otherwise imperceptible global patterns" - lead to overinterpretation of their data.

We have addressed the reviewer’s concern by modifications to the manuscript that enhance the interpretation of the findings in a clinical and biological context.

Strengths:Measurement of host immunophenotyping measures (multiparameter flow cytometry) and peripheral HIV reservoir size (IPDA) from 115 PWH on ART.Major Weaknesses:(1) Overall, there is little to no interpretability of their machine learning analyses; findings appear as a "laundry list" of parameters with no interpretation of the estimated effect size and directionality of the observed associations. For example, Figure 2 might actually give an interpretation of each X increase in immunophenotyping parameter, we saw a Y increase/decrease in HIV reservoir measure.

We have added additional text to the manuscript in which we attempt to provide more immunological and clinical interpretation of the associations. We also have emphasized that these associations are still speculative and will require additional validation. Nevertheless, our data should provide a rich source of new hypotheses regarding immune system/reservoir interaction that could be tested in future work.

(2) The correlations all appear to be relatively weak, with most Spearman R in the 0.30 range or so.

We agree with the review that the associations are mostly weak, consistent with previous studies in this area. This likely is an inherent feature of the underlying biology – the reservoir is likely associated with the immune system in complex ways and involves stochastic processes that will limit the predictability of reservoir size using any single immune parameter. We have added additional text to the manuscript to make this point clearer.

(3) The Discussion needs further work to help guide the reader. The sentence: "The correlative results from this present study corroborate many of these studies, and provide additional insights" is broad. The authors should spend some time here to clearly describe the prior literature (e.g., describe the strength and direction of the association observed in prior work linking PD-1 and HIV reservoir size, as well as specify which type of HIV reservoir measures were analyzed in these earlier studies, etc.) and how the current findings add to or are in contrast to those prior findings.

We have added additional text to the manuscript to help guide the readers through the possible biological significance of the findings and the context with respect to prior literature.

(4) The most interesting finding is buried on page 12 in the Discussion: "Uniquely, however, CD127 expression on CD4 T cells was significantly inversely associated with intact reservoir frequency." The authors should highlight this in the abstract, and title, and move this up in the Discussion. The paper describes a very high dimensional analysis and the key takeaways are not clear; the more the author can point the reader to the take-home points, the better their findings can have translatability to future follow-up mechanistic and/or validation studies.

We appreciate the reviewer’s comment. We have increased the emphasis on this finding in the revised version of the manuscript.

(5) The authors should avoid overinterpretation of these results. For example in the Discussion on page 13 "The existence of two distinct clusters of PWH with different immune features and reservoir characteristics could have important implications for HIV cure strategies - these two groups may respond differently to a given approach, and cluster membership may need to be considered to optimize a given strategy." It is highly unlikely that future studies will be performing the breadth of parameters resulting here and then use these directly for optimizing therapy.

Our analyses indicate that membership of study participants in cluster1 or cluster 2 can be fairly accurately determined by a small number of individual parameters (KLRG1 etc, Figure 4F), and measuring the cells of PWH with the degree of breadth used in this paper would not be necessary to classify PWH into these clusters. As such, we feel that it is not unrealistic to speculate that this finding could turn out to be clinically useful, if it becomes clear that the clusters are biologically meaningful.

(6) There are only TWO limitations listed here: cross-sectional study design and the use of peripheral blood samples. (The subsequent paragraph notes an additional weakness which is misclassification of intact sequences by IPDA). This is a very limited discussion and highlights the need to more critically evaluate their study for potential weaknesses.

We have expanded on the list of limitations discussed in the manuscript. In particular, we now address the size of the cohort, the composition with respect to different genders and demographics, lack of information for the timing of ART and the lack of information regarding intracellular transcriptional pathways.

(7) A major clinical predictor of HIV reservoir size and decay is the timing of ART initiation. The authors should include these (as well as other clinical covariate data - see #12 below) in their analyses and/or describe as limitations of their study.

All of the participants that make up our cohort were treated during chronic infection, and the precise timing of ART initiation is unclear in most of these cases. We have added additional information to explain this in the manuscript and include this in the list of limitations.

**Reviewer #3 (Public Review):**
Summary:This valuable study by Semenova and colleagues describes a large cross-sectional cohort of 115 individuals on ART. Participants contributed a single blood sample which underwent IPDA, and 25-color flow with various markers (pre and post-stimulation). The authors then used clustering, decision tree analyses, and machine learning to look for correlations between these immunophenotypic markers and several measures of HIV reservoir volume. They identified two distinct clusters that can be somewhat differentiated based on total HIV DNA level, intact HIV DNA level, and multiple T cell cellular markers of activation and exhaustion.The conclusions of the paper are supported by the data but the relationships between independent and dependent variables in the models are correlative with no mechanistic work to determine causality. It is unclear in most cases whether confounding variables could explain these correlations. If there is causality, then the data is not sufficient to infer directionality (ie does the immune environment impact the HIV reservoir or vice versa or both?). In addition, even with sophisticated and appropriate machine learning approaches, the models are not terribly predictive or highly correlated. For these reasons, the study is very much hypothesis-generating and will not impact cure strategies or HIV reservoir measurement strategies in the short term.

We appreciate the reviewer’s comments regarding the value of our study. We fully acknowledge that the causal nature and directionality of these associations are not yet clear and agree that the study is primarily hypothesis generating in nature. Nevertheless, we feel that the hypotheses generated will be valuable to the field. We have added additional text to the manuscript to emphasize the hypothesis generating nature of this paper.

Strengths:The study cohort is large and diverse in terms of key input variables such as age, gender, and duration of ART. Selection of immune assays is appropriate. The authors used a wide array of bioinformatic approaches to examine correlations in the data. The paper was generally well-written and appropriately referenced.Weaknesses:(1) The major limitation of this work is that it is highly exploratory and not hypothesis-driven. While some interesting correlations are identified, these are clearly hypothesis-generating based on the observational study design.

We agree that the major goal of this study was hypothesis generating and that our work is exploratory in nature. Performing experiments with mechanism testing goals in human participants with HIV is challenging. Additionally, before such mechanistic studies can be undertaken, one must have hypotheses to test. As such we feel our study will be useful for the field in helping to identify hypotheses that could potentially be tested.

(2) The study's cross-sectional nature limits the ability to make mechanistic inferences about reservoir persistence. For instance, it would be very interesting to know whether the reservoir cluster is a feature of an individual throughout ART, or whether this outcome is dynamic over time.

We agree with the reviewer’s comment. Longitudinal studies are challenging to carry out with a study cohort of this size, and addressing questions such as the one raised by the reviewer would be of great interest. We believe our study nevertheless has value in identifying hypotheses that could be tested in a longitudinal study.

(3) A fundamental issue is that I am concerned that binarizing the 3 reservoir metrics in a 50/50 fashion is for statistical convenience. First, by converting a continuous outcome into a simple binary outcome, the authors lose significant amounts of quantitative information. Second, the low and high reservoir outcomes are not actually demonstrated to be clinically meaningful: I presume that both contain many (?all) data points above levels where rebound would be expected soon after interruption of ART. Reservoir levels would also have no apparent outcome on the selection of cure approaches. Overall, dividing at the median seems biologically arbitrary to me.

The reviewer raises a valid point that the clinical significance of above or below median reservoir metrics is unclear, and that the size of the reservoir has potentially little relation to rebound and cure approaches. In the manuscript, we attempted to generate models that can predict reservoir size as a continuous variable in Figure S13 and find that this approach performs poorly, while a binarized approach was more successful. As such we have included both approaches in the manuscript. It is possible that future studies with larger sample sizes and more detailed measurements will perform better for continuous variable prediction. While this is a fairly large study (n=115) by the standards of HIV reservoir analyses, it is a small study by the standards of the machine learning field, and accurate predictive ML models for reservoir size as a continuous variable will likely require a much larger set of samples/participants. Nevertheless, we feel our work has value as a template for ML approaches that may be informative for understanding HIV/immune interactions and generates novel hypotheses that could be validated by subsequent studies.

(4) The two reservoir clusters are of potential interest as high total and intact with low % intact are discriminated somewhat by immune activation and exhaustion. This was the most interesting finding to me, but it is difficult to know whether this clustering is due to age, time on ART, other co-morbidity, ART adherence, or other possible unmeasured confounding variables.

We agree that this finding is one of the more interesting outcomes of the study. We examined a number of these variables for association with cluster membership, and these data are reported in Figure S8A-D. Age, years of ART and CD4 Nadir were all clearly different between the clusters. The striking feature of this clustering, however, is the clear separation between the two groups of participants, as opposed to a continuous gradient of phenotypes. This could reflect a bifurcation of outcomes for people with HIV, dynamic changes in the reservoir immune interactions over time, or different levels of untreated infection. It is certainly possible that some other unmeasured confounding variables contribute to this outcome and we have attempted to make this limitation clearer.

(5) At the individual level, there is substantial overlap between clusters according to total, intact, and % intact between the clusters. Therefore, the claim in the discussion that these 2 cluster phenotypes may require different therapeutic approaches seems rather speculative. That said, the discussion is very thoughtful about how these 2 clusters may develop with consideration of the initial insult of untreated infection and / or differences in immune recovery.

We agree with the reviewer that this claim is speculative, and we have attempted to moderate the language of the text in the revised version.

(6) The authors state that the machine learning algorithms allow for reasonable prediction of reservoir volume. It is subjective, but to me, 70% accuracy is very low. This is not a disappointing finding per se. The authors did their best with the available data. It is informative that the machine learning algorithms cannot reliably discriminate reservoir volume despite substantial amounts of input data. This implies that either key explanatory variables were not included in the models (such as viral genotype, host immune phenotype, and comorbidities) or that the outcome for testing the models is not meaningful (which may be possible with an arbitrary 50/50 split in the data relative to median HIV DNA volumes: see above).

We acknowledge that the predictive power of the models generated from these data is modest and we have clarified this point in the revised manuscript. As the reviewer indicates, this may result from the influence of unmeasured variables and possible stochastic processes. The data may thus demonstrate a limit to the predictability of reservoir size which may be inherent to the underlying biology. As we mention above, this study size (n-115) is fairly small for the application of ML methods, and an increased sample size will likely improve the accuracy of the models. At this stage, the models we describe are not yet useful as predictive clinical tools, but are still nonetheless useful as tools to describe the structure of the data and identify reservoir associated immune cell types.

(7) The decision tree is innovative and a useful addition, but does not provide enough discriminatory information to imply causality, mechanism, or directionality in terms of whether the immune phenotype is impacting the reservoir or vice versa or both. Tree accuracy of 80% is marginal for a decision tool.

The reviewer is correct about these points. In the revised manuscript, we have attempted to make it clear that we are not yet advocating using this approach as a decision tool, but simply a way to visualize the data and understand the structure of the dataset. As we discuss above, the models will likely need to be trained on a larger dataset and achieve higher accuracy before use as a decision tool.

(8) Figure 2: this is not a weakness of the analysis but I have a question about interpretation. If total HIV DNA is more predictive of immune phenotype than intact HIV DNA, does this potentially implicate a prior high burden of viral replication (high viral load &/or more prolonged time off ART) rather than ongoing reservoir stimulation as a contributor to immune phenotype? A similar thought could be applied to the fact that clustering could only be detected when applied to total HIV DNA-associated features. Many investigators do not consider defective HIV DNA to be "part of the reservoir" so it is interesting to speculate why these defective viruses appear to have more correlation with immunophenotype than intact viruses.

We agree with the reviewer that this observation could reflect prior viral burden and we have added additional text to make this clearer. Even so, we cannot rule out a model in which defective viral DNA is engaged in ongoing stimulation of the immune system during ART, leading to the stronger association between total DNA and the immune cell phenotypes. We hypothesize that the defective proviruses could potentially be triggering innate immune pattern recognition receptors via viral RNA or DNA, and a higher burden of the total reservoir leads to a stronger apparent association with the immune phenotype. We have included text in the discussion about this hypothesis.

(9) Overall, the authors need to do an even more careful job of emphasizing that these are all just correlations. For instance, HIV DNA cannot be proven to have a causal effect on the immunophenotype of the host with this study design. Similarly, immunophenotype may be affecting HIV DNA or the correlations between the two variables could be entirely due to a separate confounding variable

We have revised the text of the manuscript to emphasize this point, and we acknowledge that any causal relationships are, at this point, simply speculation.

(10) In general, in the intro, when the authors refer to the immune system, they do not consistently differentiate whether they are referring to the anti-HIV immune response, the reservoir itself, or both. More specifically, the sentence in the introduction listing various causes of immune activation should have citations. (To my knowledge, there is no study to date that definitively links proviral expression from reservoir cells in vivo to immune activation as it is next to impossible to remove the confounding possible imprint of previous HIV replication.) Similarly, it is worth mentioning that the depletion of intact proviruses is quite slow such that provial expression can only be stimulating the immune system at a low level. Similarly, the statement "Viral protein expression during therapy likely maintains antigen-specific cells of the adaptive immune system" seems hard to dissociate from the persistence of immune cells that were reactive to viremia.

We updated the text of the manuscript to address these points and have added additional citations as per the reviewer’s suggestion.

(11) Given the many limitations of the study design and the inability of the models to discriminate reservoir volume and phenotype, the limitations section of the discussion seems rather brief.

We have now expanded the limitations section of the discussion and added additional considerations. We now include a discussion of the study cohort size, composition and the detail provided by the assays.

**Recommendations for the authors:**

**Reviewer #1 (Recommendations For The Authors):**
A few specific comments:"This pattern is likely indicative of a more profound association of total HIV DNA with host immunophenotype relative to intact HIV DNA."Most studies I have seen (e.g. single cell from Lictherfeld/Yu group) show intact proviruses are generally more activated/detectable/susceptible to immune selection, so I have a hard time thinking defective proviruses are actually more affected by immunotype.

We hypothesize that this association is actually occurring in the opposite direction – that the defective provirus are having a greater impact on the immune phenotype, due to their greater number and potential ability to engage innate or adaptive immune receptors. We have clarified this point in the manuscript

"The existence of two distinct clusters of PWH with different immune features and reservoir characteristics could have important implications for HIV cure strategies - these two groups may respond differently to a given approach, and cluster membership may need to be considered to optimize a given strategy."I find this a bit of a reach, given that the definition of 2 categories depended on the total size.

We have modified the language of this section to reduce the level of speculation.

"This study is cross-sectional in nature and is primarily observational, so caution should be used interpreting findings associated with time on therapy".I found this an interesting statement because ultimately time on ART shows up throughout the analysis as a significant predictor, do you mean something about how time on ART could indicate other confounding variables like ART regimen or something?

We have rephrased this comment to avoid confusion. We were simply trying to make the point that we should avoid speculating about longitudinal dynamics from cross sectional data.

"As expected, the plots showed no significant correlation for intact HIV DNA versus years of ART (Figure 1B), while total reservoir size was positively correlated with the time of ART (Figure 1A, Spearman r = 0.31)."Is this expected? Studies with longitudinal data almost uniformly show intact decay, at least for the first 10 or so years of ART, and defective/total stability (or slight decay). Also probably "time on ART" to not confuse with the duration of infection before ART.

We have updated the language of this section to address this comment. We have avoided comparing our data with respect to time on ART to longitudinal studies for reasons given above.

On dimensionality reduction, as this PaCMAP seems a relatively new technique (vs tSNE and UMAP which are more standard, but absolutely have their weaknesses), it does seem important to contextualize. I think it would still be useful to show PCA and asses the % variance of each additional dimension to assess the effective dimensionality, it would be helpful to show a plot of % variance by # components to see if there is a cutoff somewhere, and if PaCMAP is really picking this up to determine the 2 dimensions/2 clusters is ideal. Figure 4B ultimately shows a lot of low/high across those clusters, and since low/high is defined categorically it's hard to know which of those dots are very close to the other categories.

We have added this analysis to the manuscript – found in Figure S9. The PCA plot indicates that members of the two clusters also separate on PCA although this separation is not as clear as for the PaCMAP plot.

Minor comments on writing etc:Intro-Needs some references on immune activation sequelae paragraph.

We have added some additional references to this section.

-"promote the entry of recently infected cells into the reservoir" -- that is only one possible mechanistic explanation, it's not unreasonable but it seems important to keep options open until we have more precise data that can illuminate the mechanism of the overabundance.

We have modified the text to discuss additional hypotheses.

-You might also reference Pankau et al Ppath for viral seeding near the time of ART.

We have added this reference.

-"Viral protein expression during therapy likely maintains antigen-specific cells of the adaptive immune system" - this was unclear to me, do you mean HIV-specific cells that act against HIV during ART? I think most studies show immunity against HIV (CD8 and CD4) wanes over time during ART.

The Goonetilleke lab has recently generated data indicating that antiviral T cell responses are remarkably stable over time on ART, but we agree with the reviewer that the idea that ongoing antigen expression in the reservoir maintains these cells is speculative. We have modified the text to make this point clearer.

-Overall I think the introduction lacked a little bit of definitional precision: i.e. is the reservoir intact vs replication competent vs all HIV DNA and whether we are talking about PWH on long-term ART and how long we should be imagining? The first years of ART are certainly different than later, in terms of dynamics. The ultimate implications are likely specific for some of these categorizations.-"persistent sequelae of the massive disruptions to T cell homeostasis and lymphoid structures that occur during untreated HIV infection" needs a lot more context/referencing. For instance, Peter Hunt showed a decrease in activation after ART a long time ago.-Heather Best et al show T cell clonality stays perturbed after ART.

We have updated the text of the introduction and added references to address the reviewer’s comments.

Results-It would be important to mention the race of participants and any information about expected clades of acquired viruses, this gets mentioned eventually with reference to the Table but the breakdown would be helpful right away.

We have added this information to the results section.

-"performed Spearman correlations", may be calculated or tested?

We have corrected the language for this sentence.

Comments on figures:-Figure 1 data on linear scale (re discussion above) -- hard to even tell if there is a decay (to match with all we know from various long-term ART studies).-Figure 4 data is shown on ln (log_e) scale, which is hard to interpret for most people.-Figures 4 C,D, and E should have box plots to visually assess the significance.-Figure 4B legend says purple/pink but I think the colors are different in the plot, could be about transparency-Figure 5 it is now not clear if log_e(?).-Figure 6 "HIV reservoir characteristics" might be better to make this more explicit. Do you mean for instance in the 6B title Total HIV DNA per million CD4+ T cells I think?

We have made these modifications.

**Reviewer #2 (Recommendations For The Authors):**
Minor Weaknesses:(1) The Introduction is too long and much of the text is not directly related to the study's research question and design.

We have streamlined the introduction in the revised manuscript.

(2) While no differences were seen by age or race, according to the authors, this is unlikely to be useful since the numbers are so small in some of these subcategories. Results from sensitivity analyses (e.g., excluding these individuals) may be more informative/useful.

We agree that the lower numbers of participants for some subgroupings makes it challenging to know for sure if there are any differences based on these variables. Have added text to clarify this. We have added age, race and gender to the LOCO analysis and to the variable inflation importance analysis (Table S5).

(3) For Figure 4, based on what was described in the Results section of the manuscript, the authors should clarify that the figures show results for TOTAL HIV DNA only (not intact DNA): "Dimension reduction machine learning approaches identified two robust clusters of PWH when using total HIV DNA reservoir-associated immune cell frequencies (Figure 4A), but not for intact or percentage intact HIV DNA (Figure 4B and 4C)".

We have added this information.

(4) The statement on page 5, first paragraph, "Interestingly, when we examined a plot of percent intact proviruses versus time on therapy (Figure 1C), we observed a biphasic decay pattern," is not new (Peluso JCI Insight 2020, Gandhi JID 2023, McMyn JCI 2023). Prior studies have clearly demonstrated this biphasic pattern and should be cited here, and the sentence should be reworded with something like "consistent with prior work", etc.

We have added citations to these studies and rephrased this comment.

(5) The Cohort and sample collection sections are somewhat thin. Further details on the cohort details should include at the very minimum some description of the timing of ART initiation (is this mostly a chronic-treated cohort?) and important covariate data such as nadir CD4+ T cell count, pre-ART viral load, duration of ART suppression, etc.

The cohort was treated during chronic infection, and we have clarified this in the manuscript. Information regarding CD4 nadir and years on ART are included in Table 1. Unfortunately, pre-ART viral load was not available for most members of this cohort, so we did not use it for analyses. The partial pre-ART viral load data is included with the dataset we are making publicly available.

**Reviewer #3 (Recommendations For The Authors):**
Minor points:(1) What is meant by CD4 nadir? Is this during primary infection or the time before ART initiation?

We have clarified this description in the manuscript. This term refers to the lowest CD4 count recorded during untreated infection.

(2) The authors claim that determinants of reservoir size are starting to emerge but other than the timing of ART, I am not sure what studies they are referring to.

We have updated the language of this section. We intended to refer to studies looking at correlates of reservoir size, and feel that this is a more appropriate term that ‘determinants’

(3) The discussion does not tie in the model-generated hypotheses with the known mechanisms that sustain the reservoir: clonal proliferation balanced by death and subset differentiation. It would be interesting to tie in the proposed reservoir clusters with these known mechanisms.

We have added additional text to the manuscript to address these mechanisms.

(4) Figure 1: Total should be listed as total HIV DNA.

We have updated this in the manuscript.

(5) Figure 1C: Worth mentioning the paper by Reeves et al which raises the possibility that the flattening of intact HIV DNA at 9 years may be spurious due to small levels of misclassification of defective as intact.

We have added this reference.

(6) "Total reservoir frequency" should be "total HIV DNA concentration"

We respectfully feel that “frequency” is a more accurate term than “concentration”, since we are expressing the reservoir as a fraction of the CD4 T cells, while “concentration” suggests a denominator of volume.

(7) Figure S2-5: label y-axis total HIV DNA.

We have updated this figure.